# PrecisionFDA Truth Challenge V2: Calling variants from short and long reads in difficult-to-map regions

## Graphical abstract

## Authors

Nathan D. Olson, Justin Wagner, Jennifer McDaniel, ..., Luoqi Chen, Fritz J. Sedlazeck, Justin M. Zook

## Correspondence

nolson@nist.gov (N.D.O.), justin.zook@nist.gov (J.M.Z.)

## In brief

Olson et al. report on the results of precisionFDA Truth Challenge V2 for variant-calling pipelines. The challenge focused on small-variant accuracy of innovative deep learning and graph-based methods, utilizing a new benchmark and with new stratifications to demonstrate strengths and weaknesses of different methods.

## Highlights

- Sixty-four submissions employing innovative variant-calling methods for three technologies

- Submissions evaluated with new GIAB benchmark sets and new genome stratifications

- Submissions differ in performance overall and in challenging genomic regions

- Challenge data are available at https://doi.org/10.18434/mds2-2336

Olson et al., 2022, Cell Genomics 2, 100129
May 11, 2022 © 2022 The Author(s).

## Resource

# PrecisionFDA Truth Challenge V2: Calling variants from short and long reads in difficult-to-map regions

Nathan D. Olson,[1],[28],* Justin Wagner,[1] Jennifer McDaniel,[1] Sarah H. Stephens,[2] Samuel T. Westreich,[3] Anish G. Prasanna,[2] Elaine Johanson,[4] Emily Boja,[4] Ezekiel J. Maier,[2] Omar Serang,[3] David Jáspez,[5] José M. Lorenzo-Salazar,[5] Adrián Muñoz-Barrera,[5] Luis A. Rubio-Rodríguez,[5] Carlos Flores,[5,6,7,8] Konstantinos Kyriakidis,[9,10] Andigoni Malousi,[10,11] Kishwar Shafin,[12] Trevor Pesout,[12] Miten Jain,[12] Benedict Paten,[12] Pi-Chuan Chang,[13] Alexey Kolesnikov,[13] Maria Nattestad,[13] Gunjan Baid,[13] Sidharth Goel,[13] Howard Yang,[13] Andrew Carroll,[13] Robert Eveleigh,[14] Mathieu Bourgey,[14] Guillaume Bourque,[14] Gen Li,[15] ChouXian Ma,[15] LinQi Tang,[15] YuanPing Du,[15] ShaoWei Zhang,[15] Jordi Morata,[16,17] Raúl Tonda,[16,17] Genís Parra,[16,17] Jean-Rémi Trotta,[16,17]

*(Author list continued on next page)*

[1]Material Measurement Laboratory, National Institute of Standards and Technology, 100 Bureau Dr, MS8312, Gaithersburg, MD 20899, USA
[2]Booz Allen Hamilton, 8283 Greensboro Drive, Mclean, VA 22102, USA
[3]DNAnexus, Inc., 1975 W El Camino Real #204, Mountain View, CA 94040, USA
[4]Office of Health Informatics, Office of the Chief Scientist, Office of the Commissioner, US Food and Drug Administration, Silver Spring, MD, USA
[5]Genomics Division, Instituto Tecnológico y de Energías Renovables (ITER), Santa Cruz de Tenerife, Spain
[6]CIBER de Enfermedades Respiratorias, Instituto de Salud Carlos III, Madrid, Spain
[7]Research Unit, Hospital Universitario N.S. de Candelaria, Santa Cruz de Tenerife, Spain
[8]Instituto de Tecnologías Biomédicas (ITB), Universidad de La Laguna, 38200 San Cristóbal de La Laguna, Spain
[9]School of Pharmacy, Aristotle University of Thessaloniki (AUTH), 541 24 Thessaloniki, Greece
[10]Genomics and Epigenomics Translational Research (GENeTres), Center for Interdisciplinary Research and Innovation, 570 01 Thessaloniki, Greece
[11]Laboratory of Biological Chemistry, School of Medicine, Aristotle University of Thessaloniki (AUTH), 541 24 Thessaloniki, Greece
[12]UC Santa Cruz Genomics Institute, University of California, Santa Cruz, 1156 High Street, Santa Cruz, CA, USA
[13]Google Inc, 1600 Amphitheater Pkwy, Mountain View, CA 94040, USA
[14]The Canadian Center for Computational Genomics (C3G), Montréal, QC, Canada
[15]HuXinDao, QingZhuHu TaiYangShan Road, KaiFu, ChangSha, HuNan, China
[16]CNAG-CRG, Centre for Genomic Regulation (CRG), Barcelona Institute of Science and Technology (BIST), Baldiri i Reixac 4, 08028 Barcelona, Spain

*(Affiliations continued on next page)*

## SUMMARY

The precisionFDA Truth Challenge V2 aimed to assess the state of the art of variant calling in challenging genomic regions. Starting with FASTQs, 20 challenge participants applied their variant-calling pipelines and submitted 64 variant call sets for one or more sequencing technologies (Illumina, PacBio HiFi, and Oxford Nanopore Technologies). Submissions were evaluated following best practices for benchmarking small variants with updated Genome in a Bottle benchmark sets and genome stratifications. Challenge submissions included numerous innovative methods, with graph-based and machine learning methods scoring best for short-read and long-read datasets, respectively. With machine learning approaches, combining multiple sequencing technologies performed particularly well. Recent developments in sequencing and variant calling have enabled benchmarking variants in challenging genomic regions, paving the way for the identification of previously unknown clinically relevant variants.

## INTRODUCTION

PrecisionFDA began in 2015 as a research effort to support the US Food and Drug Administration's (FDA's) regulatory standards development in genomics and has since expanded to support all areas of omics. The platform provides access to on-demand

high-performance computing instances, a community of experts, a library of publicly available tools, support for custom tool development, a challenge framework, and virtual shared spaces where FDA scientists and reviewers collaborate with external partners. The precisionFDA challenge framework is one of the platform's most outward-facing features. The framework enables the

**Cell Genomics**

**Resource**

Christian Brueffer,[18] Sinem Demirkaya-Budak,[19] Duygu Kabakci-Zorlu,[19] Deniz Turgut,[19] Özem Kalay,[19] Gungor Budak,[19] Kübra Narcı,[19] Elif Arslan,[19] Richard Brown,[19] Ivan J. Johnson,[19] Alexey Dolgoborodov,[19] Vladimir Semenyuk,[19] Amit Jain,[19] H. Serhat Tetikol,[19] Varun Jain,[20] Mike Ruehle,[20] Bryan Lajoie,[20] Cooper Roddey,[20] Severine Catreux,[20] Rami Mehio,[20] Mian Umair Ahsan,[21] Qian Liu,[21] Kai Wang,[21,22] Sayed Mohammad Ebrahim Sahraeian,[23] Li Tai Fang,[23] Marghoob Mohiyuddin,[23] Calvin Hung,[24] Chirag Jain,[25] Hanying Feng,[26] Zhipan Li,[26] Luoqi Chen,[26] Fritz J. Sedlazeck,[27] and Justin M. Zook[1,*]

[17]Universitat Pompeu Fabra (UPF), Barcelona, Spain
[18]Division of Oncology, Department of Clinical Sciences, Lund University, Lund, Sweden
[19]Seven Bridges Genomics, Inc, Charlestown, MA, USA
[20]Illumina, Inc., San Diego, CA, USA
[21]Raymond G. Perelman Center for Cellular and Molecular Therapeutics, Children's Hospital of Philadelphia, Philadelphia, PA 19104, USA
[22]Department of Pathology and Laboratory Medicine, Perelman School of Medicine, University of Pennsylvania, Philadelphia, PA 19104, USA
[23]Roche Sequencing Solutions, Santa Clara, CA 95050, USA
[24]WASAI Technology, Taipei, Taiwan
[25]National Human Genome Research Institute, National Institutes of Health, Bethesda, MD, USA
[26]Sentieon Inc., San Jose, CA, USA
[27]Human Genome Sequencing Center, Baylor College of Medicine, One Baylor Plaza, Houston, TX 77030, USA
[28]Lead contact
*Correspondence: nolson@nist.gov (N.D.O.), justin.zook@nist.gov (J.M.Z.)

hosting of biological data challenges in a public-facing environment, with available resources for submission testing and validation. PrecisionFDA challenges, and challenges led by other groups like DREAM (http://dreamchallenges.org.)[1–3] and Critical Assessment of Genome Interpretation,[4,5] focus experts around the world on common problems in areas of evolving science, such as genomics, proteomics, and artificial intelligence.

The first Genome In A Bottle (GIAB)-precisionFDA Truth Challenge took place in 2016, and asked participants to call small variants from short reads for two GIAB samples.[6] Benchmarks for HG001 (also called NA12878) were previously published, but no benchmarks for HG002 were publicly available at the time. This made it the first blinded germline variant-calling challenge, and the public results have been used as a point of comparison for new variant-calling methods.[7] There was no clear evidence of over-fitting methods to HG001, but performance was only assessed on relatively easy genomic regions accessible to the short reads used to form the v3.2 GIAB benchmark sets.[6]

Since the first challenge, GIAB expanded the benchmarks beyond the easy regions of the genome and improved benchmarking methods. With the advent of accurate small-variant calling from long reads using machine learning (ML),[8,9] GIAB has developed new benchmarks that cover more challenging regions of the genome,[10,11] including challenging genes that are clinically important.[12] This new small-variant benchmark (v4.2) includes SNVs and insertions or deletions (INDELs) <49 bp, integrates previously used short-read variant calls with new variant calls from 10X Genomics-linked reads and PacBio HiFi long reads, expanding the benchmark set to include 92% of the autosomes in GRCh38. This new benchmark includes difficult-to-map genes like *PMS2* and uses a local phased assembly to include highly variable genes in the major histocompatibility complex (MHC). In collaboration with the Global Alliance for Genomics and Health (GA4GH), the GIAB team defined best practices for small-variant benchmarking.[13] These best practices provide criteria for performing sophisticated variant comparisons that account for variant representation differences along with a standardized set of performance metrics. To improve insight into strengths and weaknesses of methods, for this work we developed new stratifications by genomic context (e.g., low complexity or segmental duplications). The stratified benchmarking results allow users to identify genomic regions where a particular variant-calling method performs well and where to focus optimization efforts.

In light of recent advances in genome sequencing, variant calling, and the GIAB benchmark set, we conducted a follow-up truth challenge from May to June 2020. The Truth Challenge V2 (https://precision.fda.gov/challenges/10) occurred when the v4.1 benchmark was available for HG002, but only v3.3.2 benchmark was available for HG003 and HG004. In addition to making short-read datasets available (at a lower 35× coverage than the first Truth Challenge), this challenge included long reads from two technologies to assess performance across a variety of data types. This challenge made use of the robust benchmark tools and stratifications (files with genomic coordinates for different genomic context) developed by the GA4GH Benchmarking Team and GIAB to assess performance in particularly difficult regions like segmental duplications and the MHC.[13–15] With 64 submissions across the three technologies, the results from this challenge provide a new baseline for performance to inspire ongoing advances in variant calling, particularly for challenging genomic regions.

## RESULTS

Participants were tasked with generating variant calls as Variant Call Format (VCF) files using data from one or multiple sequencing technologies for the GIAB Ashkenazi Jewish trio, available through the precisionFDA platform (Figure 1). Sequencing data were provided as FASTQ files from three technologies (Illumina, Pacific Biosciences [PacBio] HiFi, and Oxford Nanopore Technologies [ONT]) for the three human samples. The read length and coverage of the sequencing datasets were selected based on the characteristics of datasets used in practice and manufacturer recommendations (Table 1). Participants used these FASTQ files to generate variant calls against the GRCh38 version of the human reference genome.

## PrecisionFDA Truth Challenge V2

Calling Variants from Short and Long Reads in Difficult-to-Map Regions – May 1st, 2020 – June 15th, 2020

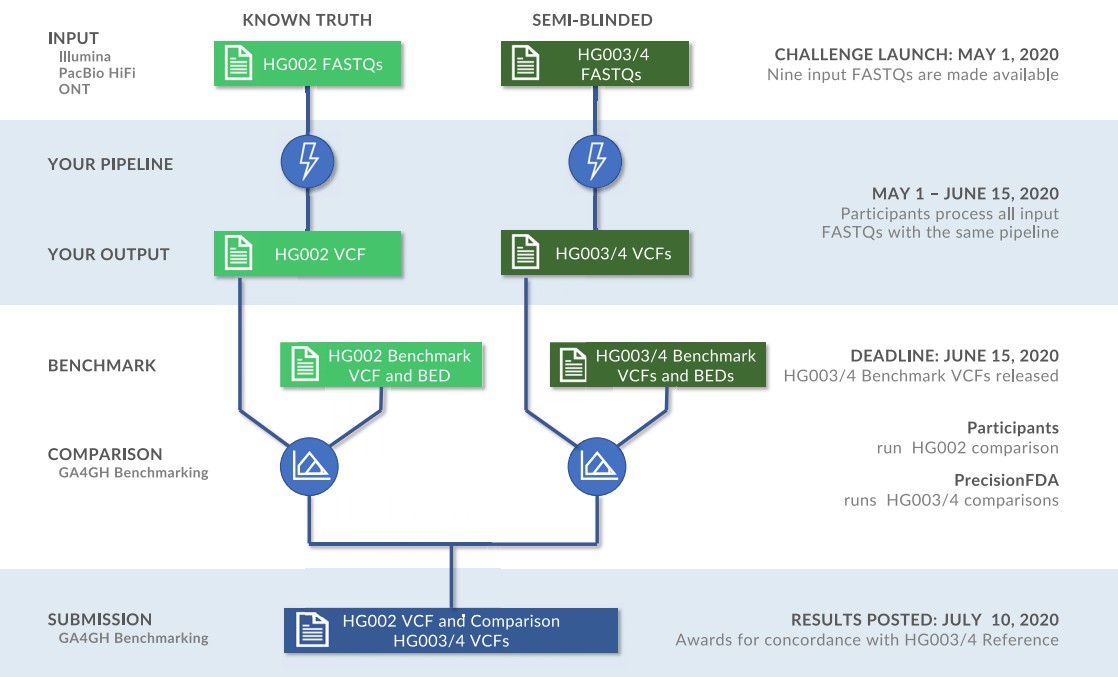

**Figure 1. Truth challenge V2 structure**

Participants were provided sequencing reads (FASTQ files) from Illumina, PacBio HiFi, and ONT for the GIAB Ashkenazi trio (HG002, HG003, and HG004). Participants uploaded VCF files for each individual before the end of the challenge, and then the new benchmarks for HG003 and HG004 were made public.

Twenty teams participated in the challenge with a total of 64 submissions, with multiple submissions from several teams. Fifteen of the 20 teams (53 of 64 submissions) volunteered to contribute to this manuscript by providing detailed methods of the pipelines they used for the challenge. The results presented here only include teams that opted to be part of this manuscript, including all the challenge winners (Figure 2A, Table S1). Thirteen of the submitted variant call sets (from teams that contributed to this manuscript) were generated using two or more sequencing technologies, Illumina, PacBio HiFi, and ONT Ultralong (see methods for datasets descriptions). For single-technology submissions, Illumina was the most common (21 out of 40), followed by PacBio (16), and ONT (3). PacBio was used in all 13 of the multiple-technology submissions, Illumina was used in all but one, and five submissions used data from all three technologies. Submissions used a variety of variant-calling methods, with most variant callers using deep-learning methods. The best-performing short-read submissions used statistical variant-calling algorithms with a graph reference rather than a standard linear reference (e.g., see DRAGEN and Seven Bridges methods in the supplemental materials). Notably, a majority of submissions used deep-learning-based variant-calling methods (Figure 2A). This was particularly true for long-read-only submissions, with 18 out of 20 using deep-learning-based methods.

Submissions were evaluated based on the harmonic mean of the parents' F1 scores for combined SNVs and INDELs. In all

benchmark regions, the top-performing submissions combined all technologies, followed by PacBio HiFi, Illumina, and ONT, with PacBio HiFi submissions having the best single-technology performance in each category (Figures 2B and 2C, Table 2). In contrast to all benchmark regions, submissions based on ONT performed better than Illumina in difficult-to-map regions despite ONT's higher indel error rate. In fact, ONT-based variant calls had slightly higher F1 scores in difficult-to-map regions than in all benchmark regions, because the benchmark for difficult-to-map regions excludes homopolymers longer than 10 bp that are called by PCR-free short reads in easy-to-map regions. The best-performing short-read call sets (DRAGEN and Seven Bridges) were statistical methods that utilized graph-based approaches, and the best-performing long-read call sets were deep-learning-based methods (DeepVariant + PEPPER, NanoCaller, Sentieon, and Roche). Performance varied substantially across stratifications, with the best-performing multi-technology call sets having similar overall performance, although with error rates that varied by a factor of 10 in the MHC. While F1 scores are similar for SNVs versus INDELs for the best-performing Illumina submissions, long-read and multi-technology submissions generally had higher F1 scores for SNVs than INDELs. ONT-based submissions had the largest decrease in performance for INDELs relative to SNVs. Submission performance for all categories (genomic regions) is provided in Table S1 with additional supplemental figures summarizing

**Table 1. Sequencing dataset characteristics**

| Technology | GIAB ID | Read length (bp) | Number of reads | Coverage |
|---|---|---|---|---|
| Illumina | HG002 | 2×151 | 415,086,209 | 35 |
| | HG003 | 2×151 | 419,192,650 | 35 |
| | HG004 | 2×151 | 420,312,085 | 35 |
| PacBio HiFi | HG002 | 12,885 | 8,449,287 | 36 |
| | HG003 | 14,763 | 7,288,357 | 35 |
| | HG004 | 15,102 | 7,089,316 | 35 |
| ONT | HG002 | 50,380 | 19,328,993 | 47 |
| | HG003 | 44,617 | 23,954,632 | 85 |
| | HG004 | 48,060 | 29,319,334 | 85 |

For read length, N50 was used to summarize PacBio and ONT read lengths; coverage was median coverage across autosomal chromosomes.

submission performance for precision (Figure S3), recall (Figure S4), and by INDEL size (Figure S5); all metrics were calculated as the harmonic mean of the parents' scores.

### Challenge highlights innovations in characterizing clinically important MHC locus

For example, recent research suggests human leukocyte antigen (HLA) types encoded in the MHC genes play a role in coronavirus disease 2019 (COVID-19) severity.[16] The MHC is a highly polymorphic ~5 Mb region of the genome that is particularly challenging for short-read methods (Figure 3). Despite difficulties associated with variant calling in this region, the Illumina graph-based pipeline developed by Seven Bridges[17] performed especially well in MHC (F1: 0.992). The Seven Bridges GRAF pipeline used in the Truth Challenge V2 utilizes a pan-genome graph that captures the genetic diversity of many populations around the world, resulting in a graph reference that accurately represents the highly polymorphic nature of the MHC region, enabling improved read alignment and variant-calling performance. The MHC region is more easily resolved with long-read-based methods as these are more likely to map in this region of high variability. The ONT-NanoCaller Medaka (F1: 0.941) ensemble submission performed well on MHC, particularly for SNVs (F1: 0.992), and is the only method that performed as well in MHC as in all genomic benchmarking regions for SNVs. In general, submissions utilizing long-read sequencing data performed better than those only using short-read data. The difference in performance between the MHC and all benchmark regions is larger for SNVs than for INDELs, and PEPPER-DV appears to have improved INDEL accuracy in the MHC, possibly because the MHC benchmark excludes some difficult homopolymers included in the all benchmark regions.

### Comparing performance for unblinded and semi-blinded samples reveals possible over-fitting of some methods

The challenge used semi-blinded samples primarily to minimize gross over-fitting of variant-calling methods to the unblinded sample. To assess potential evidence for over-fitting of methods, we explored differences in performance between the unblinded son (HG002) and semi-blinded parents' genomes (HG003 and HG004). As a metric for over-fitting, we used the error-rate ratio, defined as the ratio of 1-F1 for the parents to the son (Equation 1), such that error-rate ratios greater than one would mean that the error rate for the semi-blinded parents was higher than the error rate for the unblinded son. These error-rate ratios are likely due to a combination of factors, including differences in the sequence dataset characteristics between the three genomes, differences in the benchmark sets, and differences in participants' use of HG002 for model training and parameter optimization. The error-rate ratio was generally larger for call sets using PacBio or multiple technologies with deep learning and other ML methods compared with short-read technologies (Figure 4A). In particular, the best-performing callers had higher error-rate ratios and all used PacBio or multiple technologies with deep-learning or random forest ML methods (Figure 4B). The smaller error-rate ratios for most Illumina call sets (median 1.06, range 0.98–4.38) may relate to the maturity of short-read variant calling compared with variant calling from long reads with ML-based variant callers. For the ONT-only variant call sets, the error-rate ratio was less than 1, as the parents had higher F1 scores compared with the unblinded son (HG002). This counter-intuitive result may be caused by the parents' ONT datasets having higher coverage (85×) than the son's (47×) because ONT was not down-sampled like Illumina and PacBio (Table 1, Figure S1). The degree to which the ML models were over-fitted to the training genome (HG002) and datasets, as well as the impact of any over-fitting on variant-calling accuracy, warrants future investigation but highlights the importance of transparently describing the training and testing process, including which samples and chromosomes are used. This is particularly true given the higher degree of potential over-fitting in the best-performing long-read call sets. Note that the parents do not represent fully blinded, orthogonal samples, since HG002 shares variants with at least one of the parents, and previous benchmarks were available for the easier regions of the parents' genomes. These results highlight the need for multiple benchmark sets, sequencing datasets, and the value of established data types and variant-calling pipelines.

### Improved benchmark sets and stratifications reveal innovations in sequencing technologies and variant calling since the 2016 challenge

Since the first Truth Challenge held in 2016, variant calling, sequencing, and GIAB benchmark sets have substantially improved. The SNV error rates of the Truth Challenge V1 winners increase by as much as 10-fold when benchmarked against the new V4.2 benchmark set, compared with the V3.2 benchmark set used to evaluate the first truth challenge (Figure 4C). The V4.2 benchmark set covers 7% more of the genome than V3.2 (92% compared with 85% for HG002 on GRCh38), most importantly enabling robust performance assessment in difficult-to-map regions and the MHC.[10] The performance difference is more significant for SNVs compared with INDELs because the overall INDEL error rate is higher. Despite the higher coverage (50×) Illumina data used in the first challenge, several Illumina-only submissions from the V2 challenge performed better than the V1 challenge winners (Figure 4C). This result highlights significant

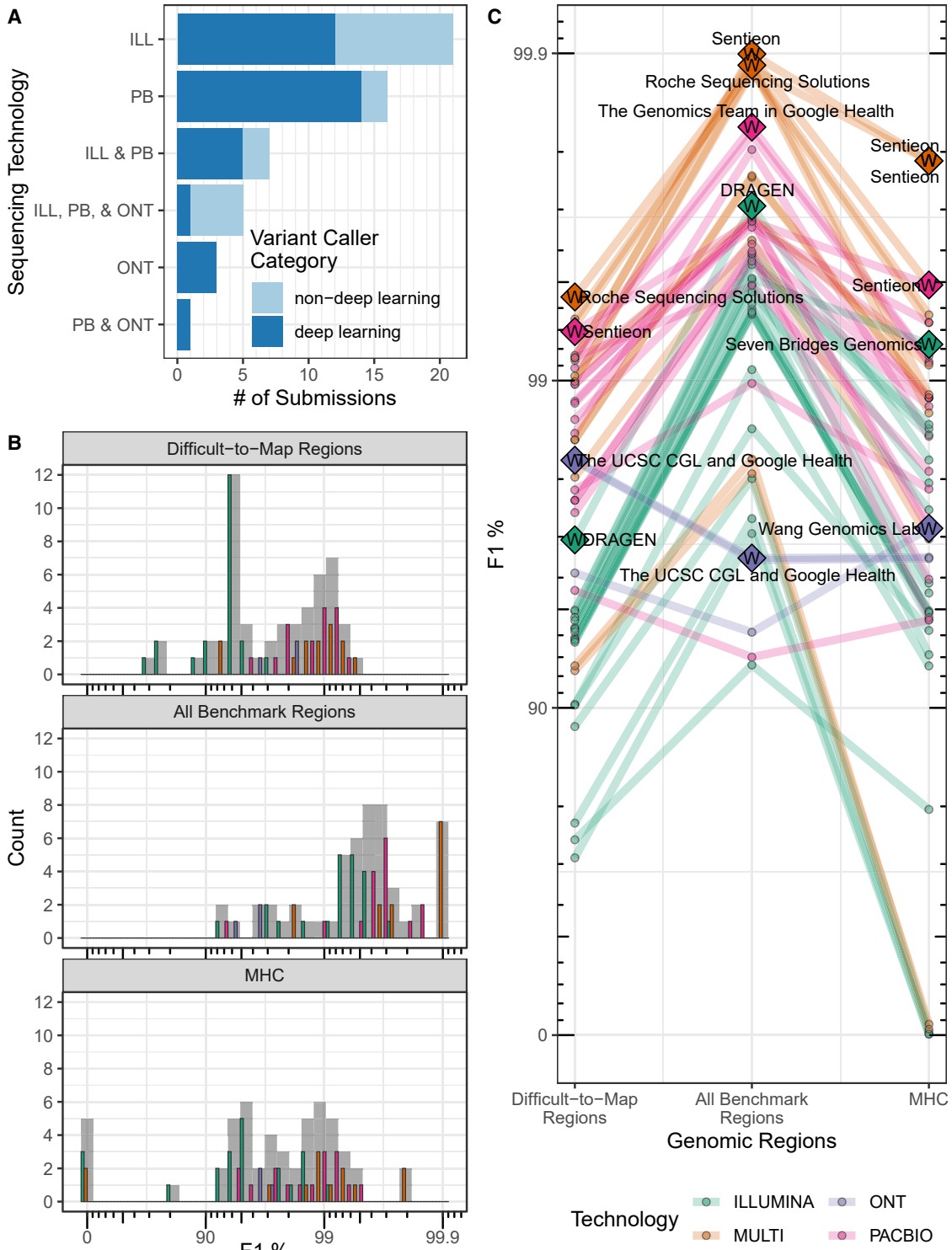

**Figure 2. Challenge submission breakdown and performance overview**

(A) Challenge submission breakdown by technology and type of variant caller used. Deep-learning methods use either a convolutional neural network or a recurrent neural network architecture for learning the variant-calling task, while non-deep-learning methods use techniques that broadly arise from statistical techniques (e.g., Bayesian and Gaussian mixture models) or other ML techniques (e.g., random forest) to differentiate variant and non-variant loci based on expert-designed features of the sequencing data.

*(legend continued on next page)*

**Table 2. Summary of challenge top performers**

| Technology | Genomic region | Participant | Performance metrics | | | F1 Rank | | |
|---|---|---|---|---|---|---|---|---|
| | | | F1 | Recall | Precision | All | Diff | MHC |
| MULTI | all[a] | Sentieon | 0.999 | 0.999 | 0.999 | 1 | 4 | 1 |
| MULTI | all[a] | Roche Sequencing Solutions | 0.999 | 0.999 | 0.999 | 1 | 1 | 7 |
| MULTI | all[a] | The Genomics Team in Google Health | 0.999 | 0.999 | 0.999 | 1 | 2 | 4 |
| MULTI | diff | Roche Sequencing Solutions | 0.994 | 0.992 | 0.996 | 1 | 1 | 7 |
| MULTI | MHC | Sentieon | 0.998 | 0.998 | 0.998 | 1 | 4 | 1 |
| ILLUMINA | all | DRAGEN | 0.997 | 0.996 | 0.998 | 1 | 1 | 5 |
| ILLUMINA | diff | DRAGEN | 0.969 | 0.961 | 0.978 | 1 | 1 | 5 |
| ILLUMINA | MHC | Seven Bridges Genomics | 0.992 | 0.989 | 0.996 | 6 | 9 | 1 |
| PACBIO | all | The Genomics Team in Google Health | 0.998 | 0.998 | 0.998 | 1 | 2 | 4 |
| PACBIO | diff | Sentieon | 0.993 | 0.991 | 0.994 | 4 | 1 | 1 |
| PACBIO | MHC | Sentieon | 0.995 | 0.993 | 0.997 | 4 | 1 | 1 |
| ONT | all | The UCSC CGL and Google Health | 0.965 | 0.947 | 0.984 | 1 | 1 | 2 |
| ONT | diff | The UCSC CGL and Google Health | 0.983 | 0.976 | 0.988 | 1 | 1 | 2 |
| ONT | MHC | Wang Genomics Lab | 0.972 | 0.964 | 0.980 | 3 | 3 | 1 |

One winner was selected for each technology/genomic region combination, and multiple winners were awarded in the case of ties. Winners were selected based on submission's F1 score for the semi-blinded samples, HG003 and HG004 (harmonic mean of the parents' F1 scores for combined SNVs and INDELs). Overall submission rank for all three genomic categories indicates submission overall performance: all, all benchmark regions; diff, difficult-to-map regions.
[a]Tie.

improvements in variant caller performance for short reads. Furthermore, advances in sequencing technologies have led to even higher accuracy, particularly in difficult-to-map regions. Improvements to the benchmarking set have allowed more accurate variant benchmarking and, in turn, facilitated advances in variant-calling methods, particularly deep-learning-based methods, which depend on the benchmark set for model training.

### Updated stratifications enable comparison of method strengths

As an example of the utility of stratifying performance in a more detailed way by genomic context with the updated stratifications, we compared the ONT PEPPER-DeepVariant (ONT-PDV) submission with the Illumina DeepVariant (Ill-DV) submission (Figure 5). The ONT-PDV submission has comparable overall performance with the Ill-DV submission for SNVs, providing an F1 of 99.64% and 99.57%, respectively, but performance differs >100-fold in some genomic contexts. Ill-DV SNV calls were more accurate in homopolymers and tandem repeats shorter than 200 bp in length. In contrast, ONT-PDV consistently had higher performance for segmental duplications, large tandem repeats, L1H, and other regions that are difficult to map with short reads. Due to the currently higher INDEL error rate for ONT R9.4 reads, Ill-DV INDEL variant calls are more accurate

for nearly every genomic context, and the F1 for INDELs in all benchmark regions was 99.59% for Ill-DV compared with 72.54% for ONT-PDV. This type of analysis can help determine the appropriate method for a desired application and understand how the strengths and limitations of technologies could be leveraged when combining technologies. High-performing multi-technology submissions successfully incorporated call sets from multiple technologies by leveraging the additional coverage and complementary strengths of different technologies.

### DISCUSSION

Public genomics community challenges, such as the precisionFDA Truth Challenges described here, provide a public baseline for independent performance evaluation at a point in time against which future methods can be compared. It is important to recognize the advancements and limitations of the benchmarks used in these challenges. For example, the GIAB V3.2 benchmark set used to evaluate the first precisionFDA Truth Challenge submissions only included the easier regions of the genome (https://precision.fda.gov/challenges/truth/results), excluding most segmental duplications and difficult-to-map regions, as well as the highly polymorphic MHC. This is shown by the fact that, when the first Truth Challenge winners were benchmarked against the new V4.2 benchmark set, which included more difficult regions of the

(B and C) Overall performance (B) and submission rank (C) varied by technology and stratification (log scale). Generally, submissions that used multiple technologies (MULTI) outperformed single-technology submissions for all three genomic context categories. (B) A histogram of F1 percentage (higher is better) for the three genomic stratifications evaluated. Submission counts across technologies are indicated by light gray bars and individual technologies by colored bars. (C) Individual submission performance. Data points represent submission performance for the three stratifications (difficult-to-map regions, all benchmark regions, MHC), and lines connect submissions. Category top performers are indicated by diamonds with Ws and labeled with team names. F1 is plotted on a phred scale with axes labels and ticks indicating F1 percentage values.

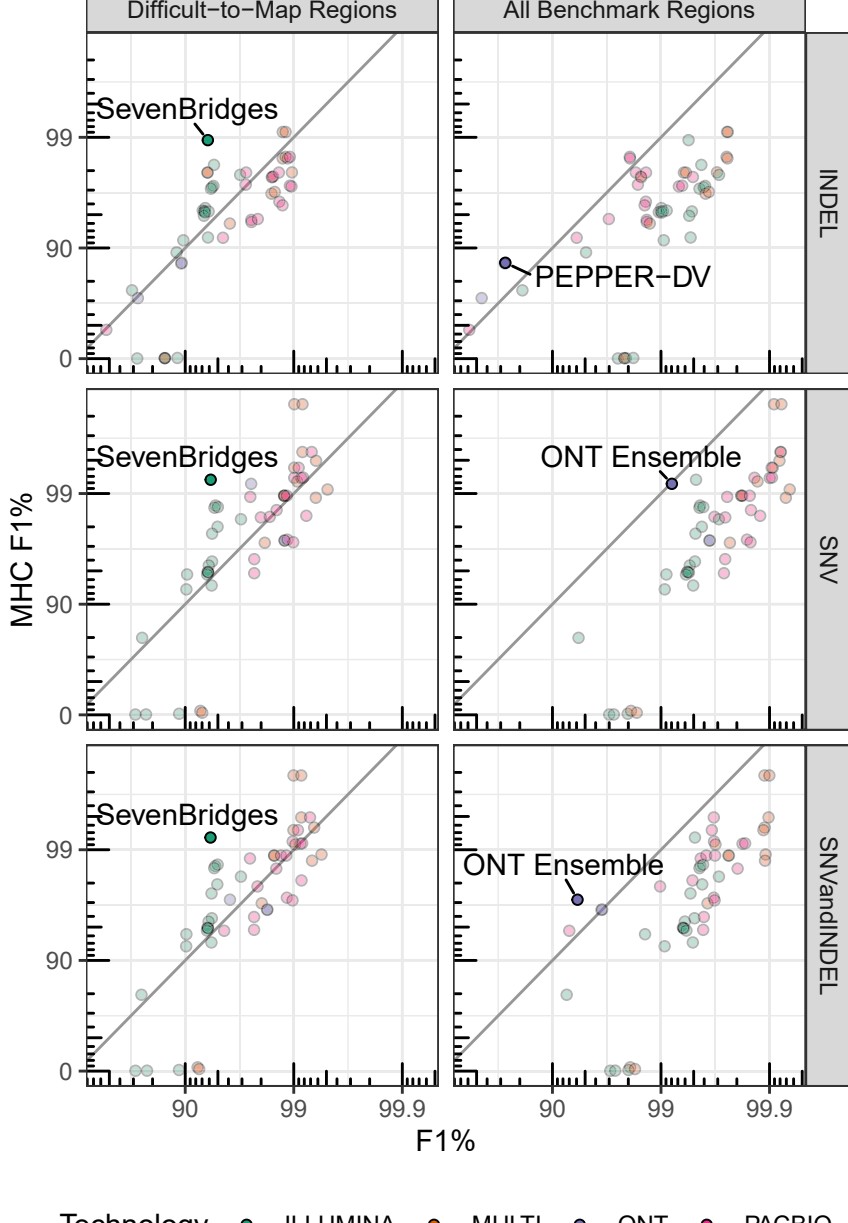

**Figure 3. Submission performance comparison for F1 metric between MHC, all benchmark regions, and difficult-to-map regions**

F1 is plotted on a phred scale with axis labels and ticks indicating F1 percentage values. Points above the diagonal black line perform better in MHC relative to all benchmark regions or the difficult-to-map regions. Submissions with the largest difference in performance between MHC and difficult-to-map or all benchmark regions for each subplot are labeled. Seven Bridges is a graph-based short-read variant caller. ONT ensemble is an ensemble of ONT variant callers; NanoCaller, Clair, and Medaka. PEPPER-DV is the ONT PEPPER-DeepVariant haplotype-aware ML variant-calling pipeline.

were difficult to map with short reads. DRAGEN's graph-based mapping method used alt-aware mapping for population haplotypes stitched into the reference with known alignments, effectively establishing alternate graph paths that reads could seed-map and align to. This reduced mapping ambiguity because reads containing population variants were attracted to the specific regions where those variants were observed.

The Seven Bridges GRAF pipeline uses a genome graph reference to map sequencing reads and uses these to genotype the sample considering the read mappings and the variant information in the graph reference. The variant calls presented in this challenge are generated using the publicly available Seven Bridges Pan-Genome GRAF Reference, constructed by augmenting the GRCh38 reference assembly with high-confidence variants selected from public databases[18–21] and also the haplotype sequences included as alternate contigs in the GRCh38 assembly relocated to their canonical positions as edges in the graph. This graph reference includes short variants as well as structural variation representing sequence diversity in the human genome (graph contains insertions of up to 9,500 base pairs, deletions spanning 580,000 base pairs, and nucleotide polymorphism spanning 4,000 base pairs). The sequence variation leads to better read mappings and variant call results, especially in highly polymorphic regions like the 4,970,558-base-pair MHC where the graph contains 71,740 nucleotide polymorphisms and 10,771 INDELs.

For the long-read methods, innovative ML-based methods were developed for this challenge. The PEPPER-DeepVariant used new approaches for selecting candidate variants and called genotypes accurately for small variants despite the relatively high error rate in raw ONT reads. Several new ML methods

genome, the performance metrics decreased as much as 10-fold (Figure 4A). It is important to note that these challenges are not just to compare and inspire new methods but to give the research and clinical sequencing community insight into what is currently possible in terms of accuracy and which methods might be applicable to the experiment in mind.

Public community challenges further help drive the methods development. A number of ground-breaking mapping + variant-calling pipelines were developed, optimized, and made available as part of this challenge. For example, the new experimental DRAGEN method used graph-based mapping and improved statistical variant-calling approaches to call variants in segmental duplications and other regions previously that

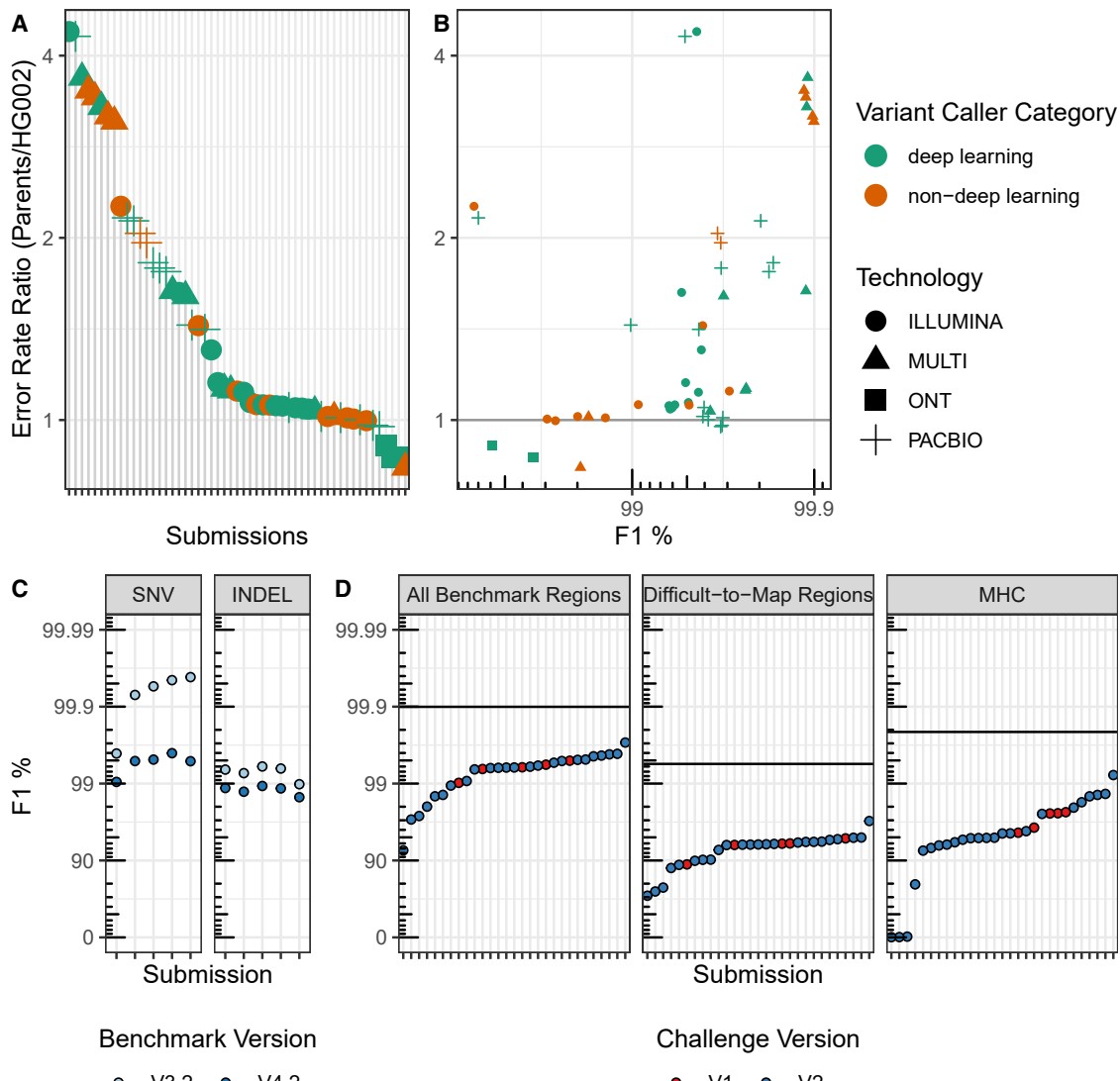

**Figure 4. Performance comparisons by sample, benchmark version, and challenges**

Ratio of error rates using semi-blinded parents' benchmark versus public son's (HG002) benchmark.

(A) Submissions ranked by error-rate ratio.

(B) Comparison of error-rate ratio with the overall performance for the parents (F1 in all benchmarking regions, as defined with Equation 1). Error rate defined as $1 - F1$. F1 is plotted on a phred scale with axis labels and ticks indicating F1 percentage values.

(C and D) Comparison of benchmarking performance for (C) different benchmark sets and (D) challenges. (C) The 2016 (V1) Truth Challenge top performers F1 performance metric for SNVs and INDELs benchmarked against the V3.2 benchmark set (used to evaluate the first challenge) and V4.2 benchmark set (used to evaluate the second challenge). Performance metrics for the same variant calls decrease substantially versus the V4.2 benchmark set because it includes more challenging regions. (D) Performance of V1 challenge top performers (using 50X Illumina sequencing) compared with V2 submissions (using only 35X Illumina sequencing) for the harmonic mean of the parents' F1 scores for combined SNVs and INDELs and the V4.2 benchmark set used to evaluate the second truth challenge. The black horizontal lines represent the performance for the overall top performer, regardless of technology used, for each stratification. For the first challenge, variant call sets for the blinded HG002 against GRCh37 were used to evaluate performance, and, for the second challenge, variant calls for the semi-blinded HG003 and HG004 against GRCh38 were used to evaluate performance. F1 is plotted on a phred scale with axes labels and ticks indicating F1 percentage values.

enabled highly accurate variant calling from the new PacBio HiFi technology. While different sequencing technologies have different strengths, robust integration of data from different technologies is challenging. Several submissions used new approaches to integrate multiple technologies and leverage the in-

dependent technology-specific information as well as additional coverage from the combining data to perform better than any individual technology.

Along with the new benchmark set and sequencing data types, we used new genomic stratifications to evaluate

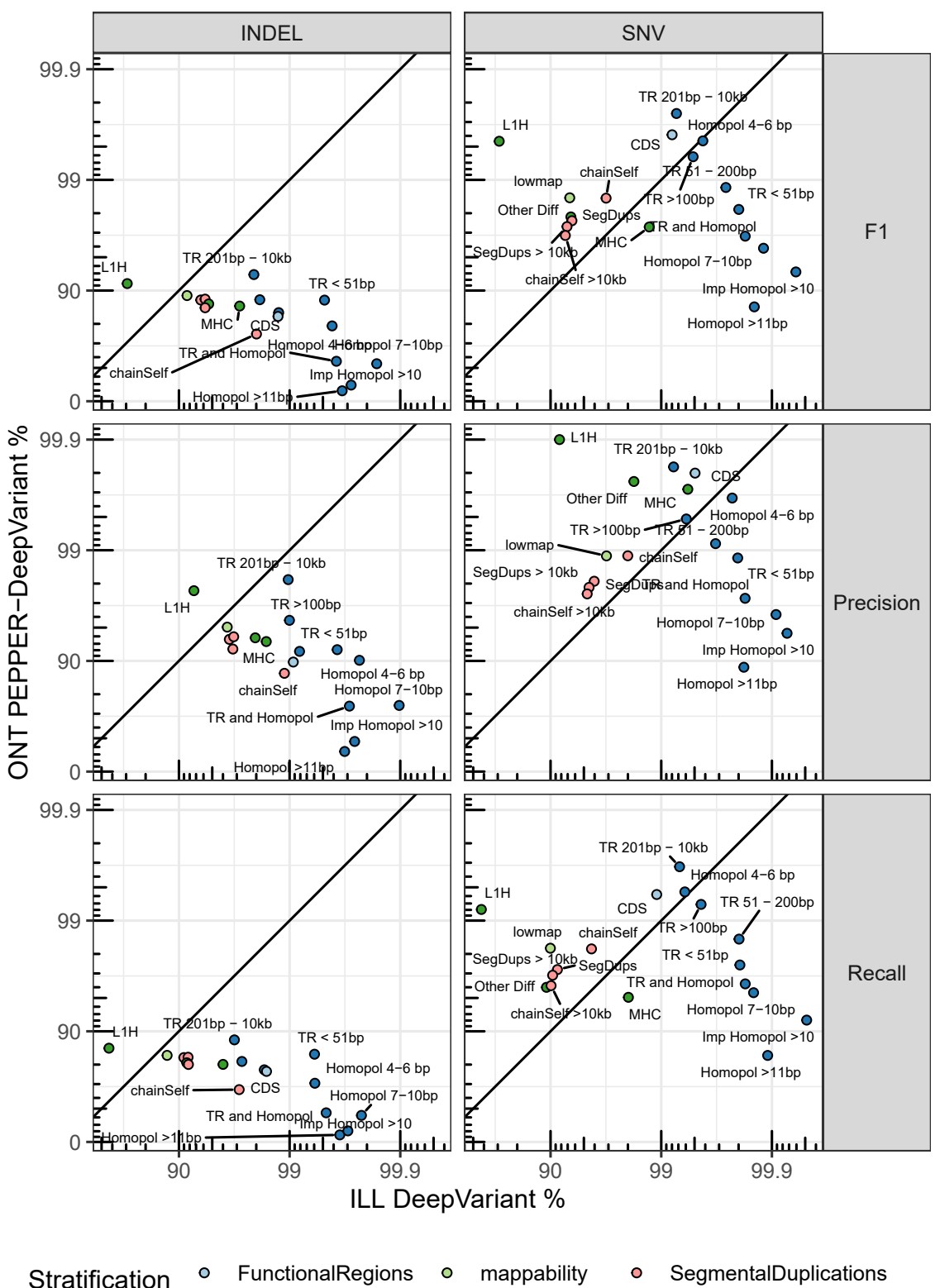

submission performance in different contexts, highlighting methods that performed best in particularly challenging regions. For example, the Seven Bridges GRAF Illumina and NanoCaller ONT submissions performed particularly well in the MHC, and the Sentieon PacBio HiFi submission performed particularly well in both the MHC and difficult-to-map regions. These submissions might have been overlooked if the performance was not stratified by context. The new stratifications presented here represent a valuable resource to the community for use in evaluating and optimizing variant-calling methods. Stratifying performance by genomic context can be valuable in at least three ways: (1) assessing the strengths and weaknesses of a method for different genome contexts and variant types, which is, for example, critical in clinical validation of bioinformatics methods;[22] (2) aiding in understanding which variants are not assessed by the benchmark; and (3) aiding in selecting the technology and bioinformatics methods that are best suited for the genomic regions of interest, e.g., MHC.

Deep learning and ML have advanced variant calling, particularly by enabling faster adoption of new sequencing technologies. In this context, care should be taken to evaluate over-training and be transparent about the data used for training, tuning, and testing. Based on results from this challenge, there is likely at least some over-fitting to training samples. Over-training can occur both to the individual (HG002) and to the properties of the sequencing runs that are used for training. Non-ML methods can also overfit, because coding and parameter selection will be guided by performance on the development set. For example, short-read variant callers that use information from long-read sequencing datasets may perform better for samples or populations included in the long-read data. Similarly, methods using graph references may perform better for samples or populations used in constructing the graph. Having clear provenance of training samples including multiple ethnicities and regions is important for the field. These results also highlight the importance of developing additional genomically diverse benchmark sets.

This challenge spurred the development and public dissemination of a diverse set of new bioinformatics methods for multiple technologies. It provides a public resource for capturing method performance at a point in time, against which future methods can be compared. New versions of these methods and new methods will continue to improve upon the methods presented here. For example, immediately after the challenge, two different participants combined the strengths of a new mapping method for long reads from one submission (winnowmap) with a new variant-calling method from another submission (PEPPER-DeepVariant) to get improved results (Figure S6).[23] The GIAB benchmarks help enable the ongoing improvements, and GIAB/GA4GH benchmarking tools enable identification of strengths and weaknesses of any method in stratified genome contexts. The new variant-calling methods presented in this challenge can help improve future versions of benchmarks that will be critical as variant-calling methods and sequencing tech-

nologies continue to improve, thus driving the advancement of research and clinical sequencing.

### Limitations of the study

Study limitations fall into two categories: those due to challenge design and limitations with voluntary participation challenges in general. For the challenge design limitations, using only samples from related individuals that share many variants resulted in challenge submissions being evaluated using semi-blinded rather than fully blinded samples. Ideally, the blinded samples would be unrelated to the unblinded sample and represent multiple ancestries. We timed this challenge to occur immediately after the release of the HG002 benchmark, and GIAB developed similar benchmarks for HG003 and HG004 during the challenge because they were the only samples for which all needed data were available. Due to the time it takes to generate benchmarks for each individual, we did not want to delay the challenge a year or more until we had benchmarks for the GIAB Han Chinese trio. Additionally, limited diversity in the GIAB samples prevented us from using fully blinded samples from multiple ancestries. (The National Institute of Standards and Technology (NIST) and the Genome in a Bottle Consortium recognize the importance of benchmarks for multiple ancestries and it is something that GIAB is actively working on to increase the diversity of the GIAB samples to understand potential effects of ancestry on accuracy.) Another practical limitation of the challenge was differences in the sequence data characteristics between individuals, particularly for the PacBio HiFi and ONT datasets. The ONT datasets had significantly higher coverage for the semi-blinded samples than the unblinded sample and semi-blinded samples. While the PacBio HiFi datasets were down-sampled to the same depth, there were differences in read-length distributions and quality scores between samples that confounded our outlier analyses. The two final limitations are related to voluntary participation challenges in general. While we strived to make our analysis of the challenge results as transparent and reproducible as possible, including making all the participant submission data publicly available, many of the participant methods are not easily reproducible and challenge submission method descriptions are inconsistent. Having fully reproducible methods for every submission would significantly increase the value of the challenge to the community. To increase challenge participation, particularly for experimental methods under active development, we did not make reproducible methods a requirement, and, while we did ask participants to provide method descriptions, they were rarely provided with the level of detail required for a peer-reviewed methods publications. Future challenges could set a higher threshold for participation regarding methods description and incentives for providing reproducible methods, although this would likely be at the cost of decreased challenge participation. Furthermore, to ensure the top submissions are reproducible, precisionFDA developers could work with challenge winners to implement their methods as apps on the precisionFDA platform. Finally, the lack of a formal

---

**Figure 5. Comparison of ONT PEPPER-DeepVariant variant call set performance with Illumina DeepVariant by genomic context**
F1 is plotted on a phred scale with axis labels and ticks indicating F1 percentage values. Points above and below the diagonal line indicate stratifications where ONT PEPPER-DeepVariant submission performance metric was higher than the Illumina DeepVariant submission. The points are colored by stratification category.

## Resource

experimental design limited our ability to make conclusive statements attributing differences in variant-calling performance to specific algorithmic characteristics and methods. A formal experimental design would significantly limit challenge participation, in turn, potentially resulting in lack of participation by developers of novel and cutting-edge methods. Challenges are designed to encourage innovative methods and provide a point of comparison for ongoing improvements, so generally do not give enduring conclusions about relative strengths of methods except at that point in time. However, such challenges provide a rich dataset for hypothesis-generating exploratory analysis.

## STAR★METHODS

Detailed methods are provided in the online version of this paper and include the following:

- KEY RESOURCES TABLE
- RESOURCE AVAILABILITY
  - Lead contact
  - Materials availability
  - Data and code availability
- EXPERIMENTAL MODEL AND SUBJECT DETAILS
  - Challenge methods
  - Challenge sequencing datasets
  - HG002 (NA24385)
  - HG003 (NA24149)
  - HG004 (NA24143)
- METHOD DETAILS
  - Challenge submission methods
  - Challenge submission evaluation
  - Genome stratifications
  - Functional regions
  - GC content
  - Genome-specific
  - Functional technically difficult
  - Low complexity
  - Other difficult
  - Segmental duplications
  - Mappability
  - Union
- QUANTIFICATION AND STATISTICAL ANALYSIS

### SUPPLEMENTAL INFORMATION

### ACKNOWLEDGMENTS

The authors would like to thank the anonymous challenge participants, as well as Drs. Megan Cleveland and Hua-Jun He for feedback on the manuscript. N.D.O. would like to thank Diana Seifert for insightful conversations and help during project development: Diana, you are missed by all who knew you. Certain commercial equipment, instruments, or materials are identified in this paper in order to specify the experimental procedure adequately. Such identification is not intended to imply recommendation or endorsement by NIST, nor is it intended to imply that the materials or equipment identified are necessarily the best available for the purpose. J.M.L.-S., A.M.-B., and L.A.R.-B. acknowledge the University of La Laguna for the training support during the PhD studies. Ministerio de Ciencia e Innovación (RTC-2017-6471-1; AEI/FEDER, UE) co-financed by the European Regional Development Funds A Way of Making Europe from the European Union, and Cabildo Insular de Tenerife supported (CGIEU0000219140) C.F.; F.J.S. was supported by NIH (UM1 HG008898); U.A., Q.L., and K.W. were supported by NIH (GM132713).

### AUTHOR CONTRIBUTIONS

Conceptualization, J.W., J.M.Z., N.D.O., E.J., E.B., S.H.S., A.G.P., E.J.M., O.S., S.T.W., and F.J.S.; data curation, N.D.O., J. McDaniel, J.W., and J.M.Z.; formal analysis – challenge participants, D.J., J.M.L.-S., A.M.-B., L.A.R.-R., C.F., K.K., A.M., K.S., T.P., M.J., B.P., P.-C.C., A.K., M.N., G. Baid, S.G., H.Y., A.C., R.E., M.B., G. Bourque, G.L., C.M., L.T., Y.D., S.Z., J. Morata, R.T., G.P., J.-R.T., C.B., S.D.-B., D.K.-Z., D.T., Ö.K., G. Budak, K.N., E.A., R.B., I.J.J., A.D., V.S., A.J., H.S.T., V.J., M.R., B.L., C.R., S.C., R.M., M.U.A., Q.L., K.W., S.M.E.S., L.T.F., M.M., C.H., C.J., H.F., Z.L., and L.C.; formal analysis – challenge results, N.D.O., J. McDaniel, J.W., and J.M.Z.; methodology, E.J., E.B., S.H.S., A.G.P., E.J.M., O.S., S.T.W., N.D.O., J.M., J.W., and J.M.Z.; project administration – challenge coordination, F.J.S., E.J., E.B., S.H.S., A.G.P., E.J.M., O.S., and S.T.W.; supervision, J.M.Z.; writing – original draft, N.D.O. and J.M.Z.; all authors reviewed and edited the manuscript.

### DECLARATION OF INTERESTS

C.B. is an employee and shareholder of SAGA Diagnostics AB. A.C., P.-C.C., A.K., M.N., G.B., S.G., and H.Y. are employees of Google, and A.C. is a shareholder. S.D.-B., D.K.-Z., D.T., Ö.K., G.B., K.N., E.A., R.B., I.J.J., A.D., V.S., A.J., and H.S.T. are employees of Seven Bridges Genomics. O.S. and S.T.W. are employees of DNAnexus. G.L., C.M., L.T.F., Y.D., and S.Z. are employees of Genetalks. V.J., M.R., B.L., C.R., S.C., and R.M. are employees of Illumina. S.M.E.S. and M.M. are employees of Roche. C.H. is an employee of Wasai Technology. H.F., Z.L., and L.C. are employees of Sentieon.

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

## STAR★METHODS

### KEY RESOURCES TABLE

| REAGENT or RESOURCE | SOURCE | IDENTIFIER |
|---|---|---|
| **Deposited data** | | |
| PacBio HiFi/CCS Sequel II | Wagner et al.[10] | SRA: SRX7083054 to SRA: SRX7083057, SRA: SRX8136474 to SRA: SRX8136477, SRA: SRX8137018 to SRA: SRX8137021 |
| Challenge sequencing datasets | | https://doi.org/10.18434/mds2-2336 |
| **Experimental models: Cell lines** | | |
| Son of Ashkenazi Jewish ancestry (HG002) | NIST Office of Reference Materials; Coriell/NIGMS; PGP | NIST RM8391/RM8392; GM24385; RRID:CVCL_1C78 |
| Father of Ashkenazi Jewish ancestry (HG003) | NIST Office of Reference Materials; Coriell/NIGMS; PGP | NIST RM8392; GM24149; RRID:CVCL_1C54 |
| Mother of Ashkenazi Jewish ancestry (HG004) | NIST Office of Reference Materials; Coriell/NIGMS; PGP | NIST RM8392; GM24143; RRID:CVCL_1C48 |
| **Software and algorithms** | | |
| hap.py | | https://github.com/Illumina/hap.py |
| seqtk | | https://github.com/lh3/seqtk |
| Code used to analyze challenge results and benchmarking results files | This paper | https://github.com/usnistgov/giab-pFDA-2nd-challenge https://doi.org/10.5281/zenodo.6384789 |
| **Other** | | |
| Sequence data, analyses, and resources related to the NIST Genome in a Bottle Consortium samples in this manuscript | This paper | https://www.nist.gov/programs-projects/genome-bottle |
| GIAB stratifications used for benchmarking | This paper | https://doi.org/10.18434/mds2-2499, also available at https://ftp-trace.ncbi.nlm.nih.gov/ReferenceSamples/giab/release/genome-stratifications/v2.0/ code to generate stratifications available at https://github.com/genome-in-a-bottle/genome-stratifications/releases/tag/v2.0 |

### RESOURCE AVAILABILITY

#### Lead contact

Further information and requests for resources and reagents should be directed to and will be fulfilled by the lead contact, Justin Zook (justin.zook@nist.gov).

#### Materials availability

DNA extracted from a single large batch of cells for the three genomes (son - HG002, father – HG003, and mother - HG004) is publicly available in National Institute of Standards and Technology Reference Materials 8391 (HG002) and 8392 (HG002-HG004), which are available at https://www.nist.gov/srm. DNA for HG002, HG003, and HG004 were extracted from publicly available cell lines GM24385 (RRID:CVCL_1C78), GM24149 (RRID:CVCL_1C54), and GM24143 (RRID:CVCL_1C48) at the Coriell Institute for Medical Research National Institute for General Medical Sciences cell line repository. The Genome in a Bottle Consortium selected these genomes for characterization as they are a trio from the Personal Genome Project that has a broader consent permitting commercial redistribution and recontacting participants for further sample collection.

#### Data and code availability

- Sequencing data are available on the precisionFDA platform and SRA, see Challenge methods and Challenge sequencing datasets for additional information and the Key resources table for SRA accession numbers. Input sequencing data, participant submitted VCFs, and benchmarking results are available at https://doi.org/10.18434/mds2-2336. Genome stratifications are

publicly available on the NIST Data Repository doi: https://doi.org/10.18434/M32190 and on the NCBI ftp site https://ftp-trace. ncbi.nlm.nih.gov/ReferenceSamples/giab/release/genome-stratifications/v2.0/
- Code used to analyze challenge results presented in the manuscript and benchmarking results files are available at https:// github.com/usnistgov/giab-pFDA-2nd-challenge and archived under doi: https://doi.org/10.5281/zenodo.6384789. The code and stratification evaluation results are in https://github.com/genome-in-a-bottle/genome-stratifications/releases/tag/ v2.0 and the NIST Data Repository https://doi.org/10.18434/M32190.
- Any additional information required to reanalyze the data reported in this paper is available from the lead contact upon request.

## EXPERIMENTAL MODEL AND SUBJECT DETAILS

### Challenge methods
The samples were sequenced under similar sequencing conditions and instruments across the three genomes. For the Illumina dataset, 2x151 bp high coverage PCR-free library was sequenced on the NovaSeq 6000 System.[24] The datasets were downsampled to 35X based on recommended coverage used in variant calling. The full 50X datasets were downsampled to 35X using seqtk (https:// github.com/lh3/seqtk) and the following command seqtk sample -s100 {fastq} 0.752733763. For PacBio HiFi, we used the library size and coverage recommended at the time by PacBio for variant calling, ~35 × 15 kb libraries. For HG002, 4 SMRT Cells were sequenced using the Sequel II System with 2.0 chemistry. Consensus basecalling was performed using the "Circular Consensus Sequencing" analysis in SMRT Link v8.0, ccs version 4.0.0. Data from the 15 kb library SMRT Cells were merged and downsampled to 35X. The combined flowcell FASTQs were downsampled using seqtk (v1.3r106, https://github.com/lh3/seqtk) to a median coverage across chromosomes 1 to 22 of 35X. Coverage was verified by mapping reads to GRCh38 using minimap2[25] and coverage was calculated with mosdepth v0.2.9[26] using a window size of 10 kb. The PacBio HiFi data are available on SRA under the following BioProjects; HG002 - PRJNA586863, HG003 - PRJNA626365, and HG004 - PRJNA626366. The ONT dataset was generated using the unsheared DNA library prep methods described in,[27] and consisted of pooled sequencing data from three PromethION R9.4 flow-cells. Basecalling was performed using Guppy Version 3.6 (https://community.nanoporetech.com). Data from three ONT PromethION flow cells were used for each of the 3 genomes, but the resulting coverage was substantially higher for the parents (85X) than the child (47X) with similar read length distributions (Figure S2).

### Challenge sequencing datasets
Links to FASTQ files provided to challenge participants on the precisionFDA platform. A free precisionFDA account is required for file access.

The sequence data are also available from the NIST data repository DOI doi: https://doi.org/10.18434/mds2-2336.

### HG002 (NA24385)
- Illumina
  - precisionFDA:
    - HG002.novaseq.pcr-free.35x.R1.fastq.gz
    - HG002.novaseq.pcr-free.35x.R2.fastq.gz
  - data.nist.gov:
    - https://opendata.nist.gov/pdrsrv/mds2-2336/input_fastqs/HG002.novaseq.pcr-free.35x.R1.fastq.gz
    - https://opendata.nist.gov/pdrsrv/mds2-2336/input_fastqs/HG002.novaseq.pcr-free.35x.R2.fastq.gz
- PacBio HiFi
  - precisionFDA: HG002_35x_PacBio_14kb-15kb.fastq.gz
  - data.nist.gov:   https://nist-midas.s3.amazonaws.com/pdrsrv/mds2-2336/input_fastqs/HG002_35x_PacBio_14kb-15kb. fastq.gz
  - SRA:
    - Bioproject: BioProject: PRJNA586863
    - Accessions: SRA: SRX7083054, SRA: SRX7083055, SRA: SRX7083056, and SRA: SRX7083057
- Oxford Nanopore
  - precisionFDA: HG002_GM24385_1_2_3_Guppy_3.6.0_prom.fastq.gz
  - data.nist.gov: https://nist-midas.s3.amazonaws.com/pdrsrv/mds2-2336/input_fastqs/HG002_GM24385_1_2_3_Guppy_ 3.6.0_prom.fastq.gz

### HG003 (NA24149)
- Illumina
  - precisionFDA:
    - HG003.novaseq.pcr-free.35x.R1.fastq.gz
    - HG003.novaseq.pcr-free.35x.R2.fastq.gz

- ○ data.nist.gov:
    - ■ https://opendata.nist.gov/pdrsrv/mds2-2336/input_fastqs/HG003.novaseq.pcr-free.35x.R1.fastq.gz
    - ■ https://opendata.nist.gov/pdrsrv/mds2-2336/input_fastqs/HG003.novaseq.pcr-free.35x.R2.fastq.gz
- ● PacBio HiFi
    - ○ precisionFDA: HG003_35x_PacBio_14kb-15kb.fastq.gz
    - ○ data.nist.gov: https://nist-midas.s3.amazonaws.com/pdrsrv/mds2-2336/input_fastqs/HG003_35x_PacBio_14kb-15kb.fastq.gz
    - ○ SRA
        - ■ Bioproject Accession: BioProject: PRJNA626365
        - ■ SRA Accessions: SRA: SRX8136474, SRA: SRX8136475, SRA: SRX8136476, and SRA: SRX8136477
- ● Oxford Nanopore
    - ○ precisionFDA: HG003_GM24149_1_2_3_Guppy_3.6.0_prom.fastq.gz
    - ○ data.nist.gov: https://nist-midas.s3.amazonaws.com/pdrsrv/mds2-2336/input_fastqs/HG003_GM24149_1_2_3_Guppy_3.6.0_prom.fastq.gz

**HG004 (NA24143)**
- ● Illumina
    - ○ precisionFDA:
        - ■ HG004.novaseq.pcr-free.35x.R1.fastq.gz
        - ■ HG004.novaseq.pcr-free.35x.R2.fastq.gz
    - ○ data.nist.gov:
        - ■ https://opendata.nist.gov/pdrsrv/mds2-2336/input_fastqs/HG004.novaseq.pcr-free.35x.R1.fastq.gz
        - ■ https://opendata.nist.gov/pdrsrv/mds2-2336/input_fastqs/HG004.novaseq.pcr-free.35x.R2.fastq.gz
- ● PacBio HiFi
    - ○ precisionFDA: HG004_35x_PacBio_14kb-15kb.fastq.gz
    - ○ data.nist.gov: https://nist-midas.s3.amazonaws.com/pdrsrv/mds2-2336/input_fastqs/HG004_35x_PacBio_14kb-15kb.fastq.gz
    - ○ SRA
        - ■ Bioproject Accession: BioProject: PRJNA626366
        - ■ SRA Accessions: SRA: SRX8137018, SRA: SRX8137019, SRA: SRX8137020, and SRA: SRX8137021
- ● Oxford Nanopore
    - ○ precisionFDA: HG004_GM24143_1_2_3_Guppy_3.6.0_prom.fastq.gz
    - ○ data.nist.gov: https://nist-midas.s3.amazonaws.com/pdrsrv/mds2-2336/input_fastqs/HG004_GM24143_1_2_3_Guppy_3.6.0_prom.fastq.gz

## METHOD DETAILS

### Challenge submission methods

Participant-provided variant calling methods are included as Table S2. Fifteen of the twenty participants, including all the challenge winners, provided methods to be made publicly available for this manuscript, a requirement for co-authorship. A random unique identifier was generated for every submission. For participants intending to remain anonymous, the unique identifier was used as the participant and submission names in the methods description.

### Challenge submission evaluation

The HG002 V4.1 benchmark set was unblinded and available to participants for model training and methods development. We used the semi-blinded HG003 and HG004 V4.2 benchmark sets to evaluate performance. The V4.1 and V4.2 benchmark sets are the latest versions of the GIAB small variant benchmark set, which utilize long- and linked-read sequencing data to expand the benchmark set into difficult regions of the genome.[10] Prior to submission, participants could benchmark their HG002 variant callsets using the precisionFDA comparator tool (https://precision.fda.gov/apps/app-F5YXbp80PBYFP059656gYxXQ-1, a free precisionFDA account is required for access). The comparator tool is an implementation of the GA4GH small variant benchmarking tool hap.py (https://github.com/Illumina/hap.py)[13] with vcfeval[14] on the precisionFDA platform. The same comparator tool was used to evaluate submission performance against the HG003 and HG004 V4.2 benchmark sets. To evaluate performance for different genomic contexts, the V2.0 genome stratifications were used (https://data.nist.gov/od/id/mds2-2190), see Genome stratifications section for a description of the different stratifications. Submissions were evaluated using the geometric mean of the HG003 and HG004 combined SNVs and INDELs F1 scores (Equation 1). We use the error rate ratio (ERR), defined as the ratio of 1-F1 for the parents (HG003 and HG004) to the son (HG002) to evaluate potential overturning.

**CellPress**

**Cell Genomics**
**Resource**

$$F1 = 2 \times (Recall \times Precision)/(Recall + Precision)$$
$$F1_{parents} = \sqrt{F1_{HG003} \times F1_{HG004}}$$
$$ERR = (1 - F1_{parents})/(1 - F1_{HG002})$$

(Equation 1)

To better understand how improvements in variant calling methods, sequencing technologies, and benchmark sets affect performance metrics, we benchmarked the first challenge winners against the updated benchmark. For the first challenge, participants submitted variant calls for HG001 and HG002 against GRCh37 using Illumina short-read sequencing data, 2x150 bp 50X coverage (higher than the more commonly used 35X in the V2 Challenge). We benchmarked the winners of the first challenge (https://precision.fda.gov/challenges/truth/results) against the V4.2 HG002 GRCh37 benchmark set. The performance metrics for the V3.2 benchmark set were obtained from the precisionFDA challenge website.

### Genome stratifications

The V2.0 genome stratifications are an update to the GA4GH genomic stratifications utilized by hap.py.[13] The V2.0 stratifications are a pared down set of stratifications with improved strata for complex regions, such as tandem repeats and segmental duplications, as well as new genome-specific stratifications for suspected copy number variants (CNVs) and known errors in the reference genome (see table below). The GRCh38 V2.0 stratifications includes 127 stratifications. The code and description of methods used to generate the stratifications and stratification evaluation results are available at https://github.com/genome-in-a-bottle/genome-stratifications/releases/tag/v2.0.

**Summary table of the V2.0 GIAB genome stratifications**

| Stratification Group | Description | # Strats | Example Stratifications | Useful for |
|---|---|---|---|---|
| FunctionalRegions | Coding regions | 2 | CDS, not in CDS | Evaluating performance in coding regions more likely to be functional |
| GC-content | Various ranges of GC-content | 14 | GC < 25%; 30% < GC < 55% | identifying GC bias in variant calling performance |
| Low Complexity | | 22 | | evaluating performance in locally repetitive, difficult to sequence contexts |
| Homopolymers | Identification of homopolymers by length | 4 | Homopolymers >101 bp; imperfect homopolymers >10 bp | evaluating performance in homopolymers, where systematic sequencing errors and complex variants frequently occur |
| Simple Repeats | Di, tri, and quad-nucleotide repeats of different lengths | 9 | Di-nucleotide repeats 11–50 bp; di-nucleotide repeats >200 bp | evaluating performance in exact Short Tandem Repeats where systematic sequencing errors and complex variants frequently occur, and variant calls are challenging if the read length is insufficient to traverse the entire repeat |
| Tandem Repeats | Tandem repeats of different lengths | 5 | Tandem repeats between 51 and 200 bp; tandem repeats >10 kb | evaluating performance in exact Short Tandem Repeats and Variable Number Tandem Repeats where systematic sequencing errors and complex variants frequently occur, and variant calls are challenging if the read length is insufficient to traverse the entire repeat |
| Other Difficult | Various difficult regions of the genome | 6 | MHC; VDJ | evaluating performance in or excluding regions where variants are difficult to call and represent due to limitations of the reference genome (e.g. gaps or errors) or being highly polymorphic in the population (MHC). |

*(Continued on next page)*

*Continued*

| Stratification Group | Description | # Strats | Example Stratifications | Useful for |
|---|---|---|---|---|
| Segmental Duplications | Segmental duplications defined using multiple methods and limited to segdups >10kb | 9 | Segdups >10 kb; selfChain | Regions with multiple similar copies in the reference, making them challenging to map and assemble. |
| Genome Specific | Difficult regions of the genome specific to one or more of the GIAB genomes. Including but not limited to complex variants, copy number variants, and structural variants. | 65 | CNVs, complex variants | evaluating performance in or excluding regions in each GIAB reference sample where small variants can be challenging to call (e.g., complex variants) or represent (e.g., CNVs and SVs) |

The updated stratification set includes the union of multiple stratifications as well as "not in" stratifications, which are useful in evaluating performance outside specific difficult genomic contexts.

The Global Alliance for Genomics and Health (GA4GH) Benchmarking Team and the Genome in a Bottle (GIAB) Consortium v2.0 stratification BED files are intended as standard resource of BED files for use in stratifying true positive, false positive, and false negative variant calls. Non-overlapping complement regions for some stratifications are also provided, as "notin" files. All stratifications that utilize the GRCh38 reference use the reference without decoy or ALT loci (ftp://ftp.ncbi.nlm.nih.gov/genomes/all/GCA/000/001/405/GCA_000001405.15_GRCh38/seqs_for_alignment_pipelines.ucsc_ids/GCA_000001405.15_GRCh38_no_alt_analysis_set.fna.gz, link checked 08/31/2020). The stratification BED files can be accessed from the NIST Public Data Repository, https://data.nist.gov/od/id/mds2-2190. Code used to generate the stratifications is available at https://github.com/genome-in-a-bottle/genome-stratifications. Genome annotation files from UCSC were used for a number of stratifications. The GRCh38 stratifications used the UCSC annotation database for the Dec. 2013 GRCh38 human genome assembly, Genome Reference Consortium Human Reference 38 GCA_000001405.15 The GRCh37 stratifications used the UCSC annotation database for the Feb. 2009 assembly of the human genome, Genome Reference

Consortium Human Reference 37 GCA_000001405.1.

### Functional regions
Two Functional Region stratifications were created to stratify variants inside and outside of coding regions. The coding regions were extracted from the RefSeq GFF file (https://ftp.ncbi.nlm.nih.gov/genomes/all/GCF/000/001/405/GCF_000001405.39_GRCh38.p13/GCF_000001405.39_GRCh38.p13_genomic.gff.gz, link checked 08/31/2020).

### GC content
Fourteen GC content stratifications were created to stratify variants into different ranges of GC content. Using the seqtk algorithm (https://github.com/lh3/seqtk, link checked 08/31/20) with the GRCh38 reference, >=x bp regions with >y% or < y% GC were identified. The output was further processed to generate 100 bp ranges of GC with an additional 50 bp slop on either side.[28]

Note that after adding 50 bp slop, 274,889 bp overlap between gc30 and gc65, or 0.05% of gc30 and 0.5% of gc65, or 0.07% of gc30 and 0.5% of gc65. The BED files with different GC ranges are almost exclusive of each other, but not completely.

We chose to stratify regions with <30% or >55% GC because these regions had decreased coverage or higher error rates for at least one of the technologies in,[28] and we added 55–60 and 60–65 because we found increased error rates in these tranches in exploratory work.

### Genome-specific
For each GIAB genome, Genome-Specific stratifications were created to identify variants in difficult regions due to potentially difficult variation in the NIST/GIAB sample, including (1) regions containing putative compound heterozygous variants, (2) regions containing multiple variants within 50 bp of each other, (3) regions with potential structural variation and copy number variation. GRCh37 stratifications were generated using vcflib vcfgeno2haplo and Unix commands to identify complex and compound variants in v3.3.2 benchmark VCF files from GIAB [6] for all samples, as well as Platinum Genomes,[29] and Real Time Genomics [30] for HG001/NA12878. To generate GRCh38 Genome Specific stratifications, the GRCh37 Genome Specific complex/compound/SVs BED files were remapped to GRCh38 using the NCBI Remapping Service (https://www.ncbi.nlm.nih.gov/genome/tools/remap). Non-overlapping complement regions for some stratifications are also provided, as "notin" files.

### Functional technically difficult
The Functional Technically Difficult stratification is used in stratifying variants by different functional, or potentially functional, regions that are also likely to be technically difficult to sequence. A list of GRCh37 difficult-to-sequence promoters, "bad promoters", was

generated from [30] supplementary file 13059_2012_3110_MOESM1_ESM.TXT (link checked 08/31/2020). The GRCh37 bad promoter-derived BED file was then remapped to GRCh38 using the NCBI remapping service (https://www.ncbi.nlm.nih.gov/genome/tools/remap).

### Low complexity

Twenty-two Low Complexity stratifications were created to identify variants in difficult regions due to different types and sizes of low complexity sequence (e.g., homopolymers, STRs, VNTRs, other locally repetitive sequences). To capture the full spectrum of repeats, we used a python script to extract Simple_repeats and Low_complexity repeats form the UCSC RepeatMasker-generated file (http://hgdownload.soe.ucsc.edu/goldenPath/hg38/database/rmsk.txt.gz, date accessed 07/22/2019)[31] and UCSC TRF-generated file (http://hgdownload.soe.ucsc.edu/goldenPath/hg38/database/simpleRepeat.txt.gz, date accessed 07/22/2019).[32]

### Other difficult

We provide nine stratifications for GRCh37 and six stratifications for GRCh38 representing additional difficult regions that do not fall into the other stratification groups. These regions include: (1) the VDJ recombination components on chromosomes 2, 14, and 22; (2) the MHC on chromosome 6; (3) L1Hs greater than 500 base pairs; (4) reference assembly contigs smaller than 500 kb; and (5) gaps in the reference assembly with 15 kb slop. In addition, we used alignments of GRCh38 to GRCh37 to identify regions that were expanded or collapsed between reference assembly releases. For GRCh37, we provide regions with alignments of either none or more than one GRCh38 contig. We also provide regions where the hs37d5 decoy sequences align to GRCh37 indicating potentially duplicated regions. We describe the identification of these regions while generating the new small variant benchmark in.[10] We generated files containing the L1H subset of LINEs greater than 500 base pairs starting with the rmsk.txt.gz file from UCSC (https://hgdownload.cse.ucsc.edu/goldenPath/hg19/database/rmsk.txt.gz) and (http://hgdownload.cse.ucsc.edu/goldenPath/hg38/database/rmsk.txt.gz) then identify entries with "L1H" and select those greater than 500 base pairs long.

### Segmental duplications

Nine Segmental Duplication stratifications were generated to identify whether variants are in segmental duplications or in regions with non-trivial self-chain alignments. Non-trivial self-chains are regions where one part of the genome aligns to another due to similarity in sequence, e.g., due to genomic duplication events. Segmental Duplications from UCSC (hgdownload.cse.ucsc.edu/goldenPath/hg38/database/genomicSuperDups.txt.gz, link checked 08/31/20) were processed to generate stratifications of all segmental duplications, segmental duplications greater than 10 kb, and regions >10 kb covered by more than 5 segmental duplications with >99% identity.[33,34] For stratifications that represent non-trivial alignments of the genome reference to itself, excluding ALT loci, the UCSC chainSelf (hgdownload.cse.ucsc.edu/goldenPath/hg38/database/chainSelf.txt.gz, link checked 08/31/20) and chainSelfLink (hgdownload.cse.ucsc.edu/goldenPath/hg38/database/chainSelfLink.txt.gz, link checked 08/31/20) were used. Together these files were used to produce stratifications for all chainSelf regions and regions greater than 10 kb.

### Mappability

Four Mappability stratifications were created to stratify variant calls based on genomic region short read mappability. Regions with low mappability for different read lengths and error rates were generated using the GEM mappability program[35] and BEDOPS genomic analysis tools.[36] Two sets of parameters were used representing low (-m 2 -e 1 -l 100) and high stringency (-m 0 -e 0 -l 250) short read mappability.

### Union

Four Union stratifications were created to identify whether variants are in, or not in, different general types of difficult regions or in any type of difficult region or complex variant. The all difficult stratification regions, is the union of all tandem repeats, all homopolymers >6 bp, all imperfect homopolymers >10 bp, all difficult to map regions, all segmental duplications, GC <25% or >65%, "Bad Promoters", and "OtherDifficultregions". Additionally stratifications are provided for the union of all difficult to map regions and all segmental duplications. For all stratifications, a "notin" non-overlapping complement is provided as "easy" regions for stratification.

### QUANTIFICATION AND STATISTICAL ANALYSIS

The input benchmarking results and code used to perform the analyses presented in this manuscript are available (https://github.com/usnistgov/giab-pFDA-2nd-challenge). The statistical programming language R was used for data analysis. Rmarkdown was used to generate individual results.[37] Packages in the Tidyverse were used for data manipulation and plotting, specifically ggplot, tidyr, and dplyr.[38]

