## [Document S2. Transparent peer review records for Olson et al. · Cell Genomics]

precisionFDA Truth Challenge V2: Calling variants from short- and long-reads in difficult-to-map regions

Nathan D. Olson^{1,27*}, Justin Wagner¹, Jennifer McDaniel¹, Sarah H. Stephens², Samuel T. Westreich³, Anish G. Prasanna², Elaine Johanson⁴, Emily Boja⁴, Ezekiel J. Maier², Omar Serang³, David Jáspez⁵, José M. Lorenzo-Salazar⁵, Adrián Muñoz-Barrera⁵, Luis A. Rubio-Rodríguez⁵, Carlos Flores^{5,6,7,8}, Konstantinos Kyriakidis^{9,10}, Andigoni Malousi^{10,11}, Kishwar Shafin¹², Trevor Pesout¹², Miten Jain¹², Benedict Paten¹², Pi-Chuan Chang¹³, Alexey Kolesnikov¹³, Maria Nattestad¹³, Gunjan Baid¹³, Sidharth Goel¹³, Howard Yang¹³, Andrew Carroll¹³, Robert Eveleigh¹⁴, Mathieu Bourgey¹⁴, Guillaume Bourque¹⁴, Gen Li¹⁵, ChouXian MA¹⁵, LinQi Tang¹⁵, YuanPing DU¹⁵, ShaoWei Zhang¹⁵, Jordi Morata¹⁶, Raúl Tonda¹⁶, Genís Parra¹⁶, Jean-Rémi Trotta¹⁶, Christian Brueffer¹⁷, Sinem Demirkaya-Budak¹⁸, Duygu Kabakci-Zorlu¹⁸, Deniz Turgut¹⁸, Özem Kalay¹⁸, Gungor Budak¹⁸, Kübra Narci¹⁸, Elif Arslan¹⁸, Richard Brown¹⁸, Ivan J Johnson¹⁸, Alexey Dolgoborodov¹⁸, Vladimir Semenyuk¹⁸, Amit Jain¹⁸, H. Serhat Tetikol¹⁸, Varun Jain¹⁹, Mike Ruehle¹⁹, Bryan Lajoie¹⁹, Cooper Roddey¹⁹, Severine Catreux¹⁹, Rami Mehio¹⁹, Mian Umair Ahsan²⁰, Qian Liu²⁰, Kai Wang^{20,21}, Sayed Mohammad Ebrahim Sahraeian²², Li Tai Fang²², Marghoob Mohiyuddin²², Calvin Hung²³, Chirag Jain²⁴, Hanying Feng²⁵, Zhipan Li²⁵, Luoqi Chen²⁵, Fritz J. Sedlazeck²⁶, Justin M. Zook^{1*}

Summary

Initial submission: Received : December 7th 2020

Scientific editor: Orli Bahcall and Emily Marcinkevicius

First round of review: Number of reviewers: 3
Revision invited: March 2nd 2021
Revision received: November 1st 2021

Second round of review: Number of reviewers: 3
Accepted: April 7th 2022

Data freely available: Yes

Code freely available: Yes

This transparent peer review record is not systematically proofread, type-set, or edited. Special characters, formatting, and equations may fail to render properly. Standard procedural text within the editor's letters has been deleted for the sake of brevity, but all official correspondence specific to the manuscript has been preserved.

Referee reports, first round of review

Reviewer #1: The paper by Olson and colleagues presents the result of the precisionFDA Truth Challenge V2. This was a competition of methods for SNP and indel callers from any combination of Illumina, PacBio HiFi, and ONT ultralong read datasets. There were three individuals (the Ashkenazi trio) sequenced and the data made available to participants. For one of the individuals, the ground truth was made available, for the others, it was hidden for the purposes of later assessment. The paper reports the results of the competition.

Challenges like these are very valuable for the community, as they help set standards and drive new method development. As such, I think the project and paper is worthy of publication, and I commend the authors for making this happen. However, I do have some concerns as outlined below.

The main major comment: For a competition such as this to be of value and not to mislead the field, it is crucial that the methods available are transparent and reproducible. These methods are currently described in a special Supplementary Material. However, this seems like someone stitched together the result of independently generated documents, without any integration or proof reading. Many of the individual sections read like a poorly written project summary from an undergrad (i.e. without complete sentences, random parentheticals, use of bullet lists instead of paragraphs). In one place (p61) there is a random yellow highlight. I can't begin to list all the problems with this document. Please modify this document so that it clearly explains each method at the standard that is expected of a methods paper.

Moreover, even after this document is better written, it would still fall short of having reproducible methods. In this day and age, it is not too much to ask the authors of each method to provide a github page for their submission with a one-touch pipeline to reproduce the results. This pipeline should not require any manual intervention, should compile, etc... Instead of a pipeline, a docker image or a conda package could also work.

Other major comments:

* In the intro, one of the highlighted contributions is the development of new stratifications of the reference genome by the genomic context (e.g. LTRs, Functional Repeats, etc). I think the paper has a good argument of why this is a valuable contribution. However, it is not thoroughly evaluated or described at the level necessary to achieve reproducibility. If this is a major contribution of the paper that is highlighted in the intro, I believe more is required. In particular,

1. There is no validation of the accuracy of this stratification. The paper simply claims that the stratification has certain properties without any validation. For example, "Low Complexity" regions are supposed to be low complexity sequences. Can the authors provide some orthogonal validation that would convince the reader that the labels corresponding to the regions are in fact accurate? If the effect of bugs (and there are always bugs) in the code that generates the stratification is significant, then there may be problems with the stratification that will propagate to all follow up work that relies on this stratification. This can be a significant setback to the field.

2. It would also greatly help if the pipeline for generating the stratification was reproducible. Currently, there are places of ambiguity. For example, "two sets of parameters were used representing low and high stringency short read mappability" -- what were the parameters. In other places, version numbers are missing (e.g. for a link to chainSelf.txt.gz on page 28, one can be more precise and quote from the file's associated readme that it annotated the Feb. 2009 assembly of the human genome). There are other examples. At the minimum, I would ask the authors to remove all places of ambiguity. Ideally, the best way would be to create a github page (or similar) with the scripts and the data that were used to generate this stratification.

* Most of the analysis focuses on the F1 score. I agree that this is a good summary statistic, but understanding the balance between precision and recall is also important. There is no breakdown of this type. I suggest adding displays such as Figure 3 and Table 2 for precision and recall.

* It would be useful to see a breakdown of accuracy as a function of indel lengths. There is currently no description of what size indels are in the benchmark and what can be reliably detected.

* I suggest adding a brief description of the benchmark used. I understand that this is a subject of its own

paper, but a paragraph or two is helpful to make this paper self contained.

Minor comments

- * I would avoid the use of a file format name ("BED") in the introduction and replace it by a description of the data type.
- * The paper uses abbreviations without first defining them (e.g. VCF, GIAB, and others)
- * Supplementary Table 1 shows up as an unformatted and unreadable glob of text in the submission. I am not sure if this is just an issue with the formatting on the submission server. If this is not done already, the table can be better formatted or provided as a CSV supplementary file.
- * In some places in the paper, the HG003 and HG004 samples are described as "blinded", but on page 10 it is argued that they are only semi-blinded. Please make the terms consistent and use the term semi-blinded everywhere in the paper. I agree with the authors that, given that these individuals are closely related to HG001, they are not "blind"
- * It says on page 11 that the SNV error rates of the V1 challenge decrease on the new benchmark. However, unless I am misunderstanding the figure or statement, Figure 6A seems to show that the F score decreases, indicating that the error rates actually increased.

Reviewer #2: The results presented in this paper are very interesting, particularly the comparisons of the strengths of different sequencing technologies and types of pipelines.

-The categorization of different pipelines (ML, graph, or statistical) is very useful, as it can elucidate the strengths and weaknesses of different variant calling techniques. However, I think this categorization could benefit from a bit more clarity and detail. It seems that the categorization is happening along (at least) two dimensions: (1) whether the pipeline uses ML techniques or not and (2) for non-ML pipelines, whether the pipeline uses a linear or graph reference. I think grouping the pipelines in this way is reasonable, but I would encourage a different choice of names, and a more detailed description of what each category entails. As I understand it, the variant callers in the "graph" and "statistical" categories are using very similar algorithmic techniques (de Bruijn graphs, HMM, Bayesian genotyping), and the major advance in the "graph" variant callers is the implementation and use of a graph reference as opposed to a linear reference, not a change in the underlying mathematical techniques, as the category names could be interpreted to imply. I think this type of confusion could be easily avoided with a short discussion of how variant calling pipelines were categorized.

- Combined SNP and INDEL, averaged, and harmonic mean F1 scores are all mentioned. Exactly what metric is being used in different parts of the paper is a little unclear. It would also be nice to see which pipelines performed better on SNPs and INDELS separately, if they were different.

- In Figure 4, I would find log ticks along the axes clearer than labeling with "log" in the axis label.

- It is stated that "SNVs were the dominant error type for Illumina sequencing". I think this is a bit misleading, as it could be incorrectly understood to mean that Illumina sequencing pipelines performed better on INDELS than on SNVs, which would be quite surprising. Instead, what I assume is meant is that because there are many more SNVs than INDELS, even though SNV error rates are lower than INDEL error rates, most errors in the Illumina based pipelines are still SNVs. I think the difference in performance between SNVs and INDELS on the different sequencing technologies could benefit from additional clearer discussion.

- In Table 2, Illumina and PacBio data looks relatively consistent between samples, but in the ONT data HG003 and HG004 have almost double the coverage as HG002. This difference is never really addressed, except as a possible explanation for higher performance in parents as compared to the son in the section on overfitting.

- The samples are sometimes referred to as HG002/3/4, and sometimes as parents/son. While some readers of the paper will likely be very familiar with the pedigree of this trio, more care could be taken to make sure things are clear to those who are not.

- In the section on overfitting, one of the possible explanations for larger performance differences in long read pipelines between the samples is given as "the Illumina datasets being more consistent in coverage and base quality across the three genomes compared to the long-read datasets". The reader is then directed to Table 1 and Figure S1. However, Table 1 doesn't show the Illumina data having particularly more consistent coverage than the PacBio data, and Figure S1 shows variability in the PacBio data base qualities distributions, but does not show Illumina data to compare to. So this is not particularly convincing evidence for the offered explanation on its own. In fact it seems possible that the larger error rate ratio seen in the long read pipelines is due to a larger fraction of the long read pipelines using ML techniques. The ML pipelines using Illumina data appear to have similar error rate ratios to those using PacBio data, for example.
- Also in the section on overfitting, there is a mention of a potential overtuning of parameters in the "statistical" variant callers. However, beyond one pipeline which appears to be a significant outlier, all of the statistical pipelines have error rate ratios very close to 1. It is not clear to me based on the data presented that there is actually any evidence of overfitting in the statistical variant callers. I think the section on overfitting is very interesting and valuable, and could benefit from slightly more detailed analysis.
- In the section comparing old to new benchmark, there is a typo saying that error rates decrease when moving from the v3.2 benchmark when in fact they increase.

Reviewer #3: The authors describe the datasets, methodology and results from the precision FDA challenge v2. The new challenge describes several improvements over the previous challenge. The gold standard now reaches into more genomic regions, including harder-to-genotype regions, and it uses a more detailed stratification of genomic regions in its analysis of results. It also includes a benchmark on HLA allele calling, which is particularly important.

The introduction is particularly sharp and persuasive. The authors do a commendable job of explaining the importance and utility of efforts like this.

Major comments:

The designations of the various approaches as "ML" "statistical" or "graph based" are questionable. I understand the desire to categorize the methods, but these categories don't make immediate sense and don't seem to be well defended anywhere in the manuscript. One problem is that the categories are not mutually exclusive (would VG followed by DeepVariant be ML or graph?), and, in particular, it is hard to draw a line between an "ML" method and a "statistical" method. Please reconsider either the categorization itself, how it's justified in the manuscript, or both.

Fig 1 refers to samples HG003 and HG004 as "blinded," but the section titled "Comparing performance for blinded and semi-blinded samples reveals possible over-tuning of some methods" refers to them as "semi-blinded," both in the text and in the captions to Figure 5 and 6. I don't understand this distinction, which seems quite important to understanding the rules and significance of the Challenge.

The section labeled "Comparing performance for blinded and semi-blinded samples reveals possible over-tuning of some methods" is quite unfocused and leaves me with many questions. Is the goal of this analysis to characterize overfitting? (Or, to say it more positively, the various methods ability to generalize beyond the first dataset?) Do the terms "over-tuning" and "overfitting" -- both used here -- mean something different? Why was this particular error ratio chosen? Should the reader draw a different conclusion about the ML-based methods versus the statistical methods? (The text is unclear on this point) Is it not concerning that the methods having the highest F1 (multi-technology ML methods) also have a very high error rate ratio (all greater than 1.5, most between 3 and 4) according to Fig 5B? I simply didn't know what to take away from this analysis.

I'll preface this next comment by saying it's a "nice-to-have" rather than a "must-have." Given that the team has made the effort to stratify genomic regions in a way that differentiates performance of the various methods ("New stratifications enable comparison of method strengths"), there would seem to be an exciting opportunity to try to establish ceiling or target for the combined methods. The idea would be to take the stratification, determine the best single-technology method in each stratum, then create a "pastiche" variant call set by simply taking all the best (per stratum) single-technology calls. If the combined-tech variant callers approach or exceed that "pastiche" call set, that's a sign that they're doing about as well as we might hope. If they're falling short, there's still room for improvement in the combined methods. To my mind, this would add a lot of impact and relevance to the work that the team has already done to identify these strata.

Minor comments:

There are two tables labeled "Table 2."

Table 1 should list SRA accessions or similar. (I noted later in the Supplement, the statement that "A free precisionFDA account is required for file access." Does this mean the datasets don't have accessions? If so, could authors explain why not? That would seem to be an important aspect of keeping these results reproducible in the long run.)

Figure 3B is quite visually confusing. I suggest replacing this with a table focusing on the winners and their F1s and *ranks* (w/r/t F1) relative to the others within each benchmark. This is obviously pretty close to what Table 2 is saying, so I'm essentially suggesting Fig 3B be merged with Table 2, expanding Table 2 with info about how each method performed and ranked in `_all_` benchmarks.

The Discussion paragraph about the graph aligners (second paragraph) is repetitive and should be revised. Two methods (DRAGEN, Seven Bridges) are described, without being compared or contrasted with each other; are they essentially the same? What's similar or different? And the discussion of the Seven Bridges method is particularly repetitive; there are a couple of sentences of that can be eliminated.

Authors' response to the first round of review

Reviewers' Comments:

Reviewer #1: The paper by Olson and colleagues presents the result of the precisionFDA Truth Challenge V2. This was a competition of methods for SNP and indel callers from any combination of Illumina, PacBio HiFi, and ONT ultralong read datasets. There were three individuals (the Ashkenazi trio) sequenced and the data made available to participants. For one of the individuals, the ground truth was made available, for the others, it was hidden for the purposes of later assessment. The paper reports the results of the competition.

Challenges like these are very valuable for the community, as they help set standards and drive new method development. As such, I think the project and paper is worthy of publication, and I commend the authors for making this happen. However, I do have some concerns as outlined below.

The main major comment: For a competition such as this to be of value and not to mislead the field, it is crucial that the methods available are transparent and reproducible. These methods are currently described in a special Supplementary Material. However, this seems like someone stitched together the result of

independently generated documents, without any integration or proof reading. Many of the individual sections read like a poorly written project summary from an undergrad (i.e. without complete sentences, random parentheticals, use of bullet lists instead of paragraphs). In one place (p61) there is a random yellow highlight. I can't begin to list all the problems with this document. Please modify this document so that it clearly explains each method at the standard that is expected of a methods paper. Moreover, even after this document is better written, it would still fall short of having reproducible methods. In this day and age, it is not too much to ask the authors of each method to provide a github page for their submission with a one-touch pipeline to reproduce the results. This pipeline should not require any manual intervention, should compile, etc... Instead of a pipeline, a docker image or a conda package could also work.

The primary goals of this manuscript are to provide a resource of benchmarking results for state-of-the-art methods at a point in time and an important example of using stratification in benchmarking. Variant callers are always evolving, so we view the benchmarking examples as the most enduring product of this manuscript. That said, we agree with the reviewer that better description of the methods is important for understanding the challenge results. Therefore, we substantially revised and reformatted the supplementary methods section, ensuring the submissions for all co-authors are appropriately documented including tools used by the participants with version info, availability, and commands used. Docker containers and conda packages are available for a majority of the methods used by challenge participants. Additionally, a few participants provided scripts used to generate submissions, which we added to the manuscript github repository, <https://github.com/usnistgov/giab-pFDA-2nd-challenge>. We particularly ensured the challenge winners' methods are all well-documented, since these are likely to be most useful to the community.

Reproducible methods were not a prerequisite for challenge participation; we were unable to provide fully reproducible methods for all challenge submissions. For example, three of the teams who participated in the challenge (10 total submissions) chose to not be included as co-authors, they did not provide methods for their submissions in the manuscript. Their results are now excluded from the manuscript figures. Additionally some of the methods are under active development or the submissions represented exploratory work.

Most importantly for the goals of this manuscript, we have ensured that the analysis of the challenge results are fully reproducible. The raw benchmarking results used to evaluate the submissions are publicly available with a permanent DOI at data.nist.gov (<https://data.nist.gov/od/id/mds2-2336>) and all the code used to produce the results in the manuscript are available at <https://github.com/usnistgov/giab-pFDA-2nd-challenge>. Furthermore the methods used to evaluate the submissions are publicly available at <https://github.com/Illumina/hap.py> as well as a bioconda package to facilitate installation, <https://anaconda.org/bioconda/hap.py>. The specific methods used for evaluations, the precisionFDA comparator tool is available on the precisionFDA platform at <https://precision.fda.gov/apps/app-F5YXbp80PBYFP059656gYxXQ-1> (a free precisionFDA account is required for access).

Other major comments:

* In the intro, one of the highlighted contributions is the development of new stratifications of the reference genome by the genomic context (e.g. LTRs, Functional Repeats, etc). I think the paper has a good argument of why this is a valuable contribution. However, it is not thoroughly evaluated or described at the level necessary to achieve reproducibility. If this is a major contribution of the paper that is highlighted in the intro, I believe more is required. In particular,

1. There is no validation of the accuracy of this stratification. The paper simply claims that the stratification has certain properties without any validation. For example, "Low Complexity" regions are supposed to be low complexity sequences. Can the authors provide some orthogonal validation that would convince the reader that the labels corresponding to the regions are in fact accurate? If the effect of bugs (and there are always bugs) in the code that generates the stratification is significant, then there may be problems with the stratification that will propagate to all follow up work that relies on this stratification. This can be a significant setback to the field.

The code used to generate the stratifications is publicly available <https://github.com/genome-in-a-bottle/genome-stratifications> and the methods are documented in the READMEs for the different stratification types. To validate the stratifications, we ran a number of sanity checks including comparing the chromosome level coverage between the two reference builds (GRCh37 and GRCh38) and evaluation of the stratification region size distributions. For file consistency, following generation of the stratified regions, all stratification files were subset to chromosomes 1-22, X and Y. It is important to note however that not all stratification types are represented in all chromosomes. The files were then sorted, merged and "N"s, specifically gaps, and pseudoautosomal regions for chromosome Y removed. Stratification files were validated by chromosome to confirm chromosome coverage between GRCh37 and GRCh38 stratifications were similar and that coverage did not exceed the chromosome length. For each chromosome within a stratification, coverage is represented by the total bases covered by the stratification relative to the chromosome reference length following removal of "N"s and pseudoautosomal regions for chromosome Y. Coverage between the references was generally consistent or as expected when differences were observed. Moreover, no coverage exceeded that of a given chromosome. We also curated a random subset of sites to help ensure they captured the intended regions. Finally we frequently use the stratifications as part of our GIAB benchmark set development and reference material characterization work, and we have found that these reliably identify areas of lower and higher performance as expected from known biases of different methods. We have clarified the availability of the code used to generate the stratifications and evaluations of the stratifications in the "Data and code availability" section: "Genome stratifications are publicly available on the NIST Data Repository [doi:10.18434/M32190](https://doi.org/10.18434/M32190) and on the NCBI ftp site <https://ftp-trace.ncbi.nlm.nih.gov/ReferenceSamples/giab/release/genome-stratifications/v2.0/>. The code and stratification evaluation results are in <https://github.com/genome-in-a-bottle/genome-stratifications/releases/tag/v2.0>."

2. It would also greatly help if the pipeline for generating the stratification was reproducible. Currently, there are places of ambiguity. For example, "two sets of parameters were used representing low and high stringency short read mappability" -- what were the parameters. In other places, version numbers are missing (e.g. for a link to chainSelf.txt.gz on page 28, one can be more precise and quote from the file's associated readme that it annotated the Feb. 2009 assembly of the human genome). There are other examples. At the minimum, I would ask the authors to remove all places of ambiguity. Ideally, the best way would be to create a github page (or similar) with the scripts and the data that were used to generate this stratification.

Thank you for pointing out the ambiguity in the methods describing the stratifications. As requested we added GEM parameters for the mappability stratifications and UCSC annotation database information. Additionally, the NIST-GIAB maintains a GitHub repository (<https://github.com/genome-in-a-bottle/genome-stratifications>) to track stratification versions and methodology for BED file generation. READMEs, associated scripts and python notebooks, if relevant, for each stratification can be found in the repository. We have included this information in the "Data and code availability" section as noted above.

* Most of the analysis focuses on the F1 score. I agree that this is a good summary statistic, but understanding the balance between precision and recall is also important. There is no breakdown of this type. I suggest adding displays such as Figure 3 and Table 2 for precision and recall.

Precision and Recall versions of Figure 3 added to supplemental figures S3 and S4 respectively. Additionally precision and recall metrics were added to Table 2.

* It would be useful to see a breakdown of accuracy as a function of indel lengths. There is currently no description of what size indels are in the benchmark and what can be reliably detected.

Added supplemental plot (Fig S6) with F1, Precision, and Recall for submissions across six INDEL size bins, deletions between 16 and 49 bp, deletions between 6 and 15 bp, deletions < 5 bp, insertions < 5bp, insertions between 6 and 15 bp, and insertions between 16 and 49 bp.

The added description of the V4.2 benchmark set in the introductions states the types of variants and size INDELS included in the benchmark.

This new small variant benchmark (v4.2), includes SNVs and INDELS < 49 bp, integrates previously-used short read variant calls with new variant calls from 10x Genomics linked reads and PacBio HiFi long reads, expanding to include 92 % of the autosomes in GRCh38. This new benchmark includes difficult-to-map genes like PMS2 and uses a local phased assembly to include highly variable genes in the MHC.

* I suggest adding a brief description of the benchmark used. I understand that this is a subject of its own paper, but a paragraph or two is helpful to make this paper self contained.

As suggested we added the following text to the introduction.

This new small variant benchmark (v4.2), includes SNVs and INDELS < 49 bp, integrates previously-used short read variant calls with new variant calls from 10x Genomics linked reads and PacBio HiFi long reads, expanding to include 92 % of the autosomes in GRCh38. This new benchmark includes difficult-to-map genes like PMS2 and uses a local phased assembly to include highly variable genes in the MHC.

Minor comments

* I would avoid the use of a file format name ("BED") in the introduction and replace it by a description of the data type.

Replaced BED files with the following text

(files with genomic coordinates for different genomic context)

* The paper uses abbreviations without first defining them (e.g. VCF, GIAB, and others)

Made sure abbreviations are defined

* Supplementary Table 1 shows up as an unformatted and unreadable glob of text in the submission. I am not sure if this is just an issue with the formatting on the submission server. If this is not done already, the table can be better formatted or provided as a CSV supplementary file.

We have simplified the table and provided it as a tsv file making the file easier to review as a plain text file. Readers can refer to the supplemental materials submission methods document for additional details regarding the submission methods.

* In some places in the paper, the HG003 and HG004 samples are described as "blinded", but on page 10 it is argued that they are only semi-blinded. Please make the terms consistent and use the term semi-blinded everywhere in the paper. I agree with the authors that, given that these individuals are closely related to HG001, they are not "blind"

Replaced blinded with semi-blinded for consistency throughout the manuscript.

* It says on page 11 that the SNV error rates of the V1 challenge decrease on the new benchmark. However, unless I am misunderstanding the figure or statement, Figure 6A seems to show that the F score decreases, indicating that the error rates actually increased.

- This was a typo and the text was corrected

Reviewer #2: The results presented in this paper are very interesting, particularly the comparisons of the strengths of different sequencing technologies and types of pipelines.

-The categorization of different pipelines (ML, graph, or statistical) is very useful, as it can elucidate the strengths and weaknesses of different variant calling techniques. However, I think this categorization could benefit from a bit more clarity and detail. It seems that the categorization is happening along (at least) two dimensions: (1) whether the pipeline uses ML techniques or not and (2) for non-ML pipelines, whether the pipeline uses a linear or graph reference. I think grouping the pipelines in this way is reasonable, but I would encourage a different choice of names, and a more detailed description of what each category entails. As I understand it, the variant callers in the "graph" and "statistical" categories are using very similar algorithmic techniques (de Bruijn graphs, HMM, Bayesian genotyping), and the major advance in the "graph" variant callers is the implementation and use of a graph reference as opposed to a linear reference, not a change in the underlying mathematical techniques, as the category names could be interpreted to imply. I think this type of confusion could be easily avoided with a short discussion of how variant calling pipelines were categorized.

Thank you for your comment. We recognize the categorizations are complicated and nuanced. We simplified the categorizations into “deep learning” and “non-deep learning”. We included the classification description in the figure 2 legend.

“Deep learning” methods use either a Convolutional Neural Network or a Recurrent Neural Network architecture for learning the variant calling task, while “non-deep learning” methods use techniques that broadly arise from statistical techniques (e.g., Bayesian and Gaussian Mixture Models) or other machine learning techniques (e.g., random forest) to differentiate variant and non-variant loci based on expert-designed features of the sequencing data.

In the results text we note that the best performing short read submissions used statistical methods but with a graph reference.

The best performing short-read submissions used statistical variant calling algorithms with a graph reference rather than a standard linear reference (e.g., see DRAGEN and Seven Bridges methods in the Supplementary Materials).

- Combined SNP and INDEL, averaged, and harmonic mean F1 scores are all mentioned. Exactly what metric is being used in different parts of the paper is a little unclear. It would also be nice to see which pipelines performed better on SNPs and INDELS separately, if they were different.

Thank you for pointing out this inconsistency describing the metric used. The metric is defined as “harmonic mean of the parents’ F1 scores for combined SNVs and INDELS.” throughout the text. We included a supplemental table with performance metrics by SNVs and INDELS as well as combined SNV and INDEL for readers who wish to further

explore the challenge results. The table is also available in the github repository with the code used to generate the figures and results presented in the manuscript, <https://github.com/usnistgov/giab-pFDA-2nd-challenge>.

- In Figure 4, I would find log ticks along the axes clearer than labeling with "log" in the axis label.

Added log ticks to all relevant axes for clarity.

- It is stated that "SNVs were the dominant error type for Illumina sequencing". I think this is a bit misleading, as it could be incorrectly understood to mean that Illumina sequencing pipelines performed better on INDELS than on SNVs, which would be quite surprising. Instead, what I assume is meant is that because there are many more SNVs than INDELS, even though SNV error rates are lower than INDEL error rates, most errors in the Illumina based pipelines are still SNVs. I think the difference in performance between SNVs and INDELS on the different sequencing technologies could benefit from additional clearer discussion.

We agree that this text implied something different from the point we were trying to make. We've revised this text to "*In general, submissions utilizing long-read sequencing data performed better than those only using short-read data. The difference in performance between the MHC and All Benchmark Regions is larger for SNVs than for INDELS, possibly because the MHC benchmark excludes some difficult homopolymers that are included in the All Benchmark Regions.*"

- In Table 2, Illumina and PacBio data looks relatively consistent between samples, but in the ONT data HG003 and HG004 have almost double the coverage as HG002. This difference is never really addressed, except as a possible explanation for higher performance in parents as compared to the son in the section on overfitting.

Our intention was to provide as much coverage to participants as possible for this new technology for small variant calling, but we acknowledge in retrospect we should have downsampled to have consistent coverage for the ONT data across genomes. We added the following text to explain the coverage differences to the end of the first paragraph of the methods section.

Data from three ONT PromethION flow cells were used for each of the 3 genomes, but the resulting coverage was substantially higher for the parents (85X) than the child (47X) with similar read length distributions (Fig. S2).

- The samples are sometimes referred to as HG002/3/4, and sometimes as parents/son. While some readers of the paper will likely be very familiar with the pedigree of this trio, more care could be taken to make sure things are clear to those who are not.

Added text to indicate parent and son sample IDs for clarity throughout the manuscript.

- In the section on overfitting, one of the possible explanations for larger performance differences in long read pipelines between the samples is given as "the Illumina datasets being more consistent in coverage and base quality across the three genomes compared to the long-read datasets". The reader is then directed to Table 1 and Figure S1. However, Table 1 doesn't show the Illumina data having particularly more consistent coverage than the PacBio data, and Figure S1 shows variability in the PacBio data base qualities distributions, but does not show Illumina data to compare to. So this is not particularly convincing evidence for the offered explanation on its own. In fact it seems possible that the larger error rate ratio seen in the long read pipelines is due to a larger fraction of the long read pipelines using ML techniques. The ML pipelines using Illumina data appear to have similar error rate ratios to those using PacBio data, for example. Also in the section on overfitting, there is a mention of a potential overtuning of parameters in the "statistical" variant callers. However, beyond one pipeline which appears to be a significant outlier, all of the statistical pipelines have error rate ratios very close to 1. It is not clear to me based on the data presented that there is actually any evidence of overfitting in the statistical variant callers. I think the section on overfitting is very interesting and valuable, and could benefit from slightly more detailed analysis.

We appreciate the reviewer's questions about this section and agree that we don't have strong evidence for what is causing the differences in over-fitting between the different methods. Unfortunately, at the time of the challenge we did not have the data needed to generate benchmarks for an unrelated sample. Despite this limitation, we determined that assessing variant calling in historically difficult regions of the genome for short read-based variant calling warranted proceeding with the challenge on the trio members. As such, the challenge design offers limited power to evaluate over-fitting, but does provide some amount of blinding to prevent gross memorization of the HG002 benchmark. That said, we agree that there are some interesting results here, which we expect will be useful to the community. We've revised the figure to remove submissions for which participants did not give methods, since their results are difficult to interpret, and this removed some of the worst outliers in over-fitting at the low end of performance. We also have clarified the classification of callers to be "deep learning" and "non-deep learning", since we found it hard to draw a line between "machine learning" and "statistical". We have revised the text to reflect what appears to be the strongest association: that all of the top-performing callsets have some evidence of over-fitting. It's difficult to know the exact reason for this and other differences from the data in this challenge, so we've revised the section to be the following:

The challenge used semi-blinded samples primarily to minimize gross over-fitting of variant calling methods to the unblinded sample. To assess potential evidence for over-fitting of methods, we explored differences in performance between the unblinded son (HG002) and semi-blinded parents' genomes (HG003 and HG004). As a metric for over-fitting, we used the error rate ratio, defined as the ratio of 1-F1 for the parents to the son (Eq. 1), such that error rate ratios greater than one mean the error rate for the semi-blinded parents was higher than the error rate for the unblinded son. These error rate ratios are likely due to a combination of factors including differences in the

sequence dataset characteristics between the three genomes, differences in the benchmark sets, and differences in participants' use of HG002 for model training and parameter optimization. The error rate ratio was generally larger for callsets using PacBio or multiple technologies with deep learning and other machine learning methods compared to short-read technologies (Fig. 5A). In particular, the best-performing callers had higher error rate ratios and all used PacBio or multiple technologies with deep learning or random forest machine learning methods (Fig. 5B). The smaller error rate ratios for most Illumina callsets (median 1.06, range 0.98 - 4.38) may relate to the maturity of short-read variant calling compared to variant calling from long reads with ML-based variant callers. For the ONT-only variant callsets, the error rate ratio was less than 1, as the parents had higher F1 scores compared to the unblinded son (HG002). This counter-intuitive result may be caused by the parents' ONT datasets having higher coverage (85X) than the son's (47X), since ONT was not downsampled like Illumina and PacBio (Table 1, Fig. S1). The degree to which the ML models were over-fitted to the training genome (HG002) and datasets as well as the impact of any over-fitting on variant calling accuracy warrants future investigation, but highlights the importance of transparently describing the training and testing process, including which samples and chromosomes are used. This is particularly true given the higher degree of potential over-fitting in the best-performing long-read callsets. Note that the parents do not represent fully blinded, orthogonal samples, since HG002 shares variants with at least one of the parents, and previous benchmarks were available for the easier regions of the parents' genomes. These results highlight the need for multiple benchmark sets, sequencing datasets, and the value of established data types and variant calling pipelines.

- In the section comparing old to new benchmark, there is a typo saying that error rates decrease when moving from the v3.2 benchmark when in fact they increase.

Thank you for pointing out this typo the text was corrected to state that there was an *increase*.

Reviewer #3: The authors describe the datasets, methodology and results from the precision FDA challenge v2. The new challenge describes several improvements over the previous challenge. The gold standard now reaches into more genomic regions, including harder-to-genotype regions, and it uses a more detailed stratification of genomic regions in its analysis of results. It also includes a benchmark on HLA allele calling, which is particularly important.

The introduction is particularly sharp and persuasive. The authors do a commendable job of explaining the importance and utility of efforts like this.

Major comments:

The designations of the various approaches as "ML" "statistical" or "graph based" are questionable. I understand the desire to categorize the methods, but these categories don't make immediate sense and don't seem to be well defended anywhere in the

manuscript. One problem is that the categories are not mutually exclusive (would VG followed by DeepVariant be ML or graph?), and, in particular, it is hard to draw a line between an "ML" method and a "statistical" method. Please reconsider either the categorization itself, how it's justified in the manuscript, or both.

Similar to reviewer 2's comment we have revised the variant categories to and provided a more detailed description of the categories. We simplified the categorizations into "deep learning" and "non-deep learning". We included the classification description in the figure 2 legend.

"Deep learning" methods use either a Convolutional Neural Network or a Recurrent Neural Network architecture for learning the variant calling task, while "non-deep learning" methods use techniques that broadly arise from statistical techniques (e.g., Bayesian and Gaussian Mixture Models) or other machine learning techniques (e.g., random forest) to differentiate variant and non-variant loci based on expert-designed features of the sequencing data.

Fig 1 refers to samples HG003 and HG004 as "blinded," but the section titled "Comparing performance for blinded and semi-blinded samples reveals possible over-tuning of some methods" refers to them as "semi-blinded," both in the text and in the captions to Figure 5 and 6. I don't understand this distinction, which seems quite important to understanding the rules and significance of the Challenge.

Replaced blinded with semi-blinded in text as appropriate for consistency

The section labeled "Comparing performance for blinded and semi-blinded samples reveals possible over-tuning of some methods" is quite unfocused and leaves me with many questions. Is the goal of this analysis to characterize overfitting? (Or, to say it more positively, the various methods ability to generalize beyond the first dataset?) Do the terms "over-tuning" and "overfitting" -- both used here -- mean something different? Why was this particular error ratio chosen? Should the reader draw a different conclusion about the ML-based methods versus the statistical methods? (The text is unclear on this point) Is it not concerning that the methods having the highest F1 (multi-technology ML methods) also have a very high error rate ratio (all greater than 1.5, most between 3 and 4) according to Fig 5B? I simply didn't know what to take away from this analysis.

As noted above in the response to Reviewer 2, we have substantially revised this section to emphasize the main points that can be drawn provisionally from the challenge as it was performed. We don't have the ability to draw strong conclusions from these data, but as suggested by Reviewer 2, we do feel these results warrant presentation, mostly to suggest the need for future work and transparency in methods. Here is the new re-written section:

The challenge used semi-blinded samples primarily to minimize gross over-fitting of variant calling methods to the unblinded sample. To assess potential evidence for over-fitting of methods, we explored differences in performance between the unblinded son (HG002) and semi-blinded parents' genomes (HG003 and HG004). As a metric for

over-fitting, we used the error rate ratio, defined as the ratio of 1-F1 for the parents to the son (Eq. 1), such that error rate ratios greater than one mean the error rate for the semi-blinded parents was higher than the error rate for the unblinded son. These error rate ratios are likely due to a combination of factors including differences in the sequence dataset characteristics between the three genomes, differences in the benchmark sets, and differences in participants' use of HG002 for model training and parameter optimization. The error rate ratio was generally larger for callsets using PacBio or multiple technologies with deep learning and other machine learning methods compared to short-read technologies (Fig. 5A). In particular, the best-performing callers had higher error rate ratios and all used PacBio or multiple technologies with deep learning or random forest machine learning methods (Fig. 5B). The smaller error rate ratios for most Illumina callsets (median 1.06, range 0.98 - 4.38) may relate to the maturity of short-read variant calling compared to variant calling from long reads with ML-based variant callers. For the ONT-only variant callsets, the error rate ratio was less than 1, as the parents had higher F1 scores compared to the unblinded son (HG002). This counter-intuitive result may be caused by the parents' ONT datasets having higher coverage (85X) than the son's (47X), since ONT was not downsampled like Illumina and PacBio (Table 1, Fig. S1). The degree to which the ML models were over-fitted to the training genome (HG002) and datasets as well as the impact of any over-fitting on variant calling accuracy warrants future investigation, but highlights the importance of transparently describing the training and testing process, including which samples and chromosomes are used. This is particularly true given the higher degree of potential over-fitting in the best-performing long-read callsets. Note that the parents do not represent fully blinded, orthogonal samples, since HG002 shares variants with at least one of the parents, and previous benchmarks were available for the easier regions of the parents' genomes. These results highlight the need for multiple benchmark sets, sequencing datasets, and the value of established data types and variant calling pipelines.

I'll preface this next comment by saying it's a "nice-to-have" rather than a "must-have." Given that the team has made the effort to stratify genomic regions in a way that differentiates performance of the various methods ("New stratifications enable comparison of method strengths"), there would seem to be an exciting opportunity to try to establish a ceiling or target for the combined methods. The idea would be to take the stratification, determine the best single-technology method in each stratum, then create a "pastiche" variant call set by simply taking all the best (per stratum) single-technology calls. If the combined-tech variant callers approach or exceed that "pastiche" call set, that's a sign that they're doing about as well as we might hope. If they're falling short, there's still room for improvement in the combined methods. To my mind, this would add a lot of impact and relevance to the work that the team has already done to identify these strata.

We recognize the reviewer's desire for a combined callset based on the combination of variant callers by stratification. Some participants in fact used this approach for their submissions, for example a few submissions used different variant calling methods for the MHC region than the rest of the genome or used one variant callset for SNVs and

another for INDELS. However, it is difficult to robustly combine callsets due to complex interactions between stratifications and differences in variant representation between variant callers. We hope to work on benchmarking methods that account for interactions between stratifications to better characterize variant caller performance which could inform a “pastiche” variant callset described by the reviewer.

Minor comments:

There are two tables labeled "Table 2."

Tables renumbers and references to tables updated in text.

Table 1 should list SRA accessions or similar. (I noted later in the Supplement, the statement that "A free precisionFDA account is required for file access." Does this mean the datasets don't have accessions? If so, could authors explain why not? That would seem to be an important aspect of keeping these results reproducible in the long run.)

The fastqs, participant vcf files, and benchmarking results are available on nist.data.gov, <https://doi.org/10.18434/mds2-2336>. The PacBio data used in this challenge are downsampled from a larger data set, the SRA accessions are provided in the Supplemental Material for the full dataset

Figure 3B is quite visually confusing. I suggest replacing this with a table focusing on the winners and their F1s and *ranks* (w/r/t F1) relative to the others within each benchmark. This is obviously pretty close to what Table 2 is saying, so I'm essentially suggesting Fig 3B be merged with Table 2, expanding Table 2 with info about how each method performed and ranked in `_all_` benchmarks.

Thank you for this suggestion, we added ranks based on F1 metrics for all three challenge categories to Table 2, to highlight challenge winners overall performance. We added the full challenge results table to the supplemental material for readers who want to dig deeper into the challenge results.

/*

The Discussion paragraph about the graph aligners (second paragraph) is repetitive and should be revised. Two methods (DRAGEN, Seven Bridges) are described, without being compared or contrasted with each other; are they essentially the same? What's similar or different? And the discussion of the Seven Bridges method is particularly repetitive; there are a couple of sentences of that can be eliminated.

The following text was added to the discussion to clarify the differences between the two graph based variant calling pipelines.

Seven Bridges GRAF pipeline uses a genome graph reference to map sequencing reads, and uses these to genotype the sample taking into account the read mappings and the variant information in the graph reference. The variant calls presented in this challenge are generated using the publicly available Seven Bridges' Pan-

GenomeGlobal GRAF Reference, constructed by augmenting the GRCh38 reference assembly with high-confidence variants selected from public databases (1000 Genomes, Mills' INDELS, Simons Diversity Project, gnomAD), and also the haplotype sequences included as alternate contigs in the GRCh38 assembly relocated to their canonical positions as edges in the graph. This graph reference includes short variants as well as structural variation representing sequence diversity in the human genome (graph contains Insertions of up to 9500 base-pairs, deletions spanning 580,000 base-pairs, and nucleotide polymorphism spanning 4,000 base-pairs). The sequence variation leads to better read mappings and variant call results, especially in highly polymorphic regions like the 4,970,558 base-pair MHC where the graph contains 71,740 nucleotide polymorphisms and 10,771 INDELS.

Referees' report, second round of review

Reviewer #1: I thank the reviewers for addressing some of my comments, but my major issue remains unaddressed. The authors claim that the primary goal of this manuscript is to provide a benchmarking resource. I had thought that the benchmark was the subject of a separate paper? But even if it were not, the manuscript I am reviewing DOES have a focus on the methods, rather than the benchmark only. For example, the abstract talks about how "new methods out-performed..." and how "challenge submissions included a number of innovative methods...". There are whole sections dedicated to analyzing the submitted methods (e.g. "Comparing performance for unblinded and semi-blinded samples reveals possible over-fitting of some methods").

I appreciate that reproducible methods were not a prerequisite for this challenge and that even so, some methods are reproducible. Nevertheless, I think it is inappropriate to publish the results of even a single method that is not reproducible. The precision/recall numbers published in this paper will de facto become a standard which other methods will need to beat. To ensure that this standard is legitimate, methods that generate it must be reproducible. This has been a community standard for many (I'd estimate around 10) years.

Since the authors are unwilling to make the submitted methods reproducible, I recommend a rejection. I did not look at the other issues as they are moot if the submitted methods are not reproducible. I think publishing a major challenge paper like this with non-reproducible submissions will do significant damage to the field, as has happened with previous challenges that did not have reproducible methods (e.g. Alignathon).

Reviewer #2: Thank you for the revisions to the original manuscript. My comments on the original manuscript were largely well addressed. However, I still think there could be a little more exposition on SNV and INDEL performance separately. The inclusion of the table of raw results at the end of the paper is appreciated, as is the sharing of the results in a github repo. It would be nice to point readers to the github repo (I didn't see a mention of it, though I may have missed it). I also think that even just a short qualitative mention of SNV vs INDEL performance would be good to see (even just something like "The best performing submissions tended to perform best on both SNVs or INDELS", or "Long read based submissions were best on INDELS while short read base submissions were best on SNVs", depending on what the results actually were). I think these types of high level observations would be useful to many readers who might not have the time or inclination to go into the data to make them themselves.

Other than that, my only comment is that there are a few spots that indicate a close proof read might be helpful to pick out any remaining areas where the writing can be cleaned up a little. For example, at the end of page 6 into page 7 the statement about how performance was measured is repeated multiple times in the same paragraph, which makes it not flow particularly well. And in the "Challenge Highlights

Innovations in Characterizing Clinically-important MHC" section, the first sentence is a run-on. I think in general the manuscript is well written and easy to understand, but a small amount of editorial cleanup could be helpful.

Reviewer #3: The authors have addressed the critiques from the first round quite well. I have a lingering concern that there's little for the reader to take away from the discussion of overfitting, and that the paper might be improved from omitting this section. But overall the paper is strong enough to publish.

Authors' response to the second round of review

Reviewers' Comments:

Reviewer #1: I thank the reviewers for addressing some of my comments, but my major issue remains unaddressed. The authors claim that the primary goal of this manuscript is to provide a benchmarking resource. I had thought that the benchmark was the subject of a separate paper? But even if it were not, the manuscript I am reviewing DOES have a focus on the methods, rather than the benchmark only. For example, the abstract talks about how "new methods out-performed..." and how "challenge submissions included a number of innovative methods...". There are whole sections dedicated to analyzing the submitted methods (e.g. "Comparing performance for unblinded and semi-blinded samples reveals possible over-fitting of some methods").

I appreciate that reproducible methods were not a prerequisite for this challenge and that even so, some methods are reproducible. Nevertheless, I think it is inappropriate to publish the results of even a single method that is not reproducible. The precision/recall numbers published in this paper will de facto become a standard which other methods will need to beat. To ensure that this standard is legitimate, methods that generate it must be reproducible. This has been a community standard for many (I'd estimate around 10) years.

Since the authors are unwilling to make the submitted methods reproducible, I recommend a rejection. I did not look at the other issues as they are moot if the submitted methods are not reproducible. I think publishing a major challenge paper like this with non-reproducible submissions will do significant damage to the field, as has happened with previous challenges that did not have reproducible methods (e.g. Alignathon).

We appreciate the reviewer's emphasis on reproducibility. To this end we have set up a github repository (<https://github.com/usnistgov/giab-pFDA-2nd-challenge>) with the code used to analyze the challenge results, including all results presented in the manuscript. The challenge submission VCFs, and benchmarking results along with the sequence datasets provided to the challenge participants are archived on data.nist.gov (<https://data.nist.gov/od/id/mds2-2336>) providing a DOI for the data ensuring long term accessibility. It is unfortunate that a number of the challenge participants did not provide sufficient documentation so that their variant calling methods are reproducible. This was a limitation of the challenge design as the participants as reproducible methods was not a requirement for challenge participation. (There is a trade off between requirements for participation, we did not want to make challenge participation overly burdensome, in turn limiting the number of submissions.) However, the top submissions, which we expect will be most useful to the community, are well documented. Furthermore, the variant calling methods are advancing rapidly as new versions of the methods used by some of the top submissions have already been released. The focus of this work is to provide a snapshot into how the top variant calling

methods are performing at this point in time and inform readers about the resources for small variant benchmarking that have been developed by the Genome and a Bottle team.

We added the following text to the next *Study Limitations* section noting that not all of the participants provide sufficient information for fully reproducible methods and that this was a limitation of the challenge design and that future challenges should have a higher threshold for participation regarding methods description and incentives for providing reproducible methods. Furthermore, to ensure the top submissions are reproducible the pFDA developers could work with challenge winners to implement their methods as apps on the pFDA platform.

“The two final limitations are related to voluntary participation challenges in general. While we strived to make our analysis of the challenge results as transparent and reproducible as possible, including making all the participant submission data publicly available. Many of the participant methods are not easily reproducible and challenge submission method descriptions are inconsistent. Having fully reproducible methods for every submission would significantly increase the value of the challenge to the community. To increase challenge participation, particularly for experimental methods under active development, we did not make reproducible methods a requirement and while we did ask participants to provide method descriptions, they were rarely provided with the level of detail required for a peer-reviewed methods publications. Future challenges could set a higher threshold for participation regarding methods description and incentives for providing reproducible methods, although this would likely be at the cost of decreased challenge participation. Furthermore, to ensure the top submissions are reproducible precisionFDA developers could work with challenge winners to implement their methods as apps on the pFDA platform.”

Reviewer #2: Thank you for the revisions to the original manuscript. My comments on the original manuscript were largely well addressed. However, I still think there could be a little more exposition on SNV and INDEL performance separately.

The inclusion of the table of raw results at the end of the paper is appreciated, as is the sharing of the results in a github repo. It would be nice to point readers to the github repo (I didn't see a mention of it, though I may have missed it).

A link to the github repository is included in the data availability and methods sections. Additionally the following sentence was added to the manuscript summary so that the challenge resources are easier to find.

The pFDA challenge results can be found at <https://precision.fda.gov/challenges/10>, the challenge data are available on the precisionFDA platform and archived under [doi:10.18434/mds2-2336](https://doi.org/10.18434/mds2-2336), finally the code use to generate the analyses presented in the manuscript are available at <https://github.com/usnistgov/giab-pFDA-2nd-challenge>.

I also think that even just a short qualitative mention of SNV vs INDEL performance would be good to see (even just something like "The best performing submissions tended to perform best on both SNVs or INDELS", or "Long read based submissions were best on INDELS while short read base submissions were best on SNVs", depending on what the results actually were). I think these types of high level observations would be useful to many readers who might not have the time or inclination to go into the data to make them themselves.

We agree that there are interesting differences between SNV and INDEL performance, so we've added the following text "While F1 scores are similar for SNVs vs. INDELS for the best-performing Illumina submissions, long-read and multi-technology submissions generally had higher F1 scores for SNVs than INDELS. ONT-based submissions had the largest difference in performance between SNVs and INDELS."

Other than that, my only comment is that there are a few spots that indicate a close proof read might be helpful to pick out any remaining areas where the writing can be cleaned up a little. For example, at the end of page 6 into page 7 the statement about how performance was measured is repeated multiple times in the same paragraph, which makes it not flow particularly well. And in the "Challenge Highlights Innovations in Characterizing Clinically-important MHC" section, the first sentence is a run-on. I think in general the manuscript is well written and easy to understand, but a small amount of editorial cleanup could be helpful.

Text was cleaned up and identified errors were corrected.

Reviewer #3: The authors have addressed the critiques from the first round quite well. I have a lingering concern that there's little for the reader to take away from the discussion of overfitting, and that the paper might be improved from omitting this section. But overall the paper is strong enough to publish.

Thank you for your review. We agree the overfitting section is the weakest in the manuscript. Cell Genomics does not have a supplemental results section to emphasize the limitations of the overfitting analyses; we included the following text in the new study limitations section.

Relevant Limitations of Study Text

For the challenge design limitations, using only samples from related individuals that share many variants resulted in challenge submissions being evaluated using semi-blinded rather than fully-blinded samples. Ideally the blinded samples would be unrelated to the unblinded sample and represent multiple ancestries. While ideal, this was not practical for we timed this challenge to occur immediately after the release of the HG002 benchmark, and GIAB developed similar benchmarks for HG003 and HG004 during the challenge because they were the only samples for which all needed data were available. due to the time it takes to generate benchmarks for each individual, we did not want to delay the challenge a year or more until we had benchmarks for the GIAB Han Chinese trio. Additionally, limited diversity in the Genome In A Bottle samples prevented us from using fully blinded samples from multiple ancestries. (NIST and the Genome In A Bottle consortium recognizes the importance of benchmarks for multiple ancestries and it is something that GIAB is actively working on increasing the diversity of the GIAB samples to understand potential effects of ancestry on accuracy.) Another practical limitation of the challenge was differences in the sequence data characteristics between individuals, particularly for the PacBio HiFi and ONT datasets. The ONT datasets had significantly higher coverage for the semi-blinded samples than the unblinded sample. and semi-blinded samples and while the PacBio HiFi datasets were down sampled to the same depth there were differences in read length distributions and quality scores between samples that confounded our outlier analyses.

Final editorial correspondence with Reviewer #1:**Editor, sharing the response (see above) from authors to Reviewer #1 –**

In the last round of review, you had indicated concerns about the issue of reproducibility in the Truth Challenge results. The authors have worked to provide a response, below, outlining their efforts to address the concern and to be upfront about the limitations that the Challenge design ended up having in this regard.

Editorially, and grounded in the support of two other reviewers, we are interested in publishing this study and feel that reporting these findings will be valuable to the community, so long as the accompanying limitations have been appropriately discussed. Prior to final acceptance of the paper I wanted to share the authors' responses with you so that you know that they have taken your comments seriously, that we have similarly editorially considered the issue, and also to invite you to share any follow-up remarks, suggestions for further discussion or caveating that you think should be included in the paper, or anything else that you'd like to add.

Reviewer #1 response -

Thank you for letting me know. I am glad that you and the authors appreciated the importance of my concerns, and I appreciate how forthright the limitations discussion is about the issue. In terms of the justifications the authors give, I do appreciate that having the bar of reproducibility would make such a project more challenging and lower participation. But that is the price of having such a standard. For example, one could similarly argue that requiring any methods paper to be reproducible will make publishing harder and keep some otherwise great ideas from being published. Still, it is an important standard that has been set and is generally enforced by the community, because we believe that the benefits of such a standard are worth the costs. As I mentioned, publishing such a non-reproducible paper can do damage to the field, as has happened with previous challenges that did not have reproducible methods (e.g. Alignathon). I do hope that the benefits of this paper will outweigh this potential problem.